# Functional implications of MIR domains in protein *O*-mannosylation

**Antonella Chiapparino[1†], Antonija Grbavac[2†], Hendrik RA Jonker[3], Yvonne Hackmann[1], Sofia Mortensen[1], Ewa Zatorska[2], Andrea Schott[2], Gunter Stier[1], Krishna Saxena[3], Klemens Wild[1], Harald Schwalbe[3], Sabine Strahl[2]*, Irmgard Sinning[1]***

[1]Heidelberg University Biochemistry Center (BZH), Heidelberg, Germany; [2]Centre for Organismal Studies (COS), Heidelberg University, Heidelberg, Germany; [3]Institute for Organic Chemistry and Chemical Biology, Center for Biomolecular Magnetic Resonance (BMRZ), Goethe University, Frankfurt am Main, Germany

**Abstract** Protein *O*-mannosyltransferases (PMTs) represent a conserved family of multispanning endoplasmic reticulum membrane proteins involved in glycosylation of S/T-rich protein substrates and unfolded proteins. PMTs work as dimers and contain a luminal MIR domain with a β-trefoil fold, which is susceptive for missense mutations causing α-dystroglycanopathies in humans. Here, we analyze PMT-MIR domains by an integrated structural biology approach using X-ray crystallography and NMR spectroscopy and evaluate their role in PMT function in vivo. We determine Pmt2- and Pmt3-MIR domain structures and identify two conserved mannose-binding sites, which are consistent with general β-trefoil carbohydrate-binding sites (α, β), and also a unique PMT2-subfamily exposed FKR motif. We show that conserved residues in site α influence enzyme processivity of the Pmt1-Pmt2 heterodimer in vivo. Integration of the data into the context of a Pmt1-Pmt2 structure and comparison with homologous β-trefoil – carbohydrate complexes allows for a functional description of MIR domains in protein *O*-mannosylation.

**\*For correspondence:**
sabine.strahl@cos.uni-heidelberg.
de (SS);
irmi.sinning@bzh.uni-heidelberg.
de (IS)

[†]These authors contributed equally to this work

**Competing interests:** The authors declare that no competing interests exist.

## Introduction

Protein *O*-mannosyltransferases (PMTs) of the PMT family are multispanning membrane glycosyltransferases (GT family 39; http://www.cazy.org) of the GT-C fold (*Albuquerque-Wendt et al., 2019*; *Bai et al., 2019*) that catalyze the transfer of mannose from dolichol monophosphate-activated mannose (Dol-P-Man) to the hydroxyl group of serine and threonine residues of proteins in the endoplasmic reticulum (ER) (reviewed in *Neubert and Strahl, 2016*). PMT-based *O*-mannosylation is an evolutionarily conserved and vital process, key for cell wall integrity and protein quality control in fungi (*Arroyo et al., 2011*; *Gentzsch and Tanner, 1996*; *Xu et al., 2013*). In humans, impairment of this classic type of *O*-mannosylation leads to a series of neuromuscular disorders, due to α-dystroglycan hypo-glycosylation, known as α−dystroglycanopathies (reviewed in *Endo, 2015*).

   PMTs are divided into three subfamilies (PMT1, PMT2, and PMT4) (reviewed in *Neubert and Strahl, 2016*) and form functional dimers either within or across subfamilies (*Akasaka-Manya et al., 2006*; *Girrbach and Strahl, 2003*; *Figure 1A*). Mammals have only two PMT proteins, POMT1 and POMT2, working as a heterodimer (*Akasaka-Manya et al., 2006*). Conversely, *Saccharomyces cerevisiae* has seven different PMT family members, six of which have confirmed enzymatic activity (Pmt1-Pmt6) (*Gentzsch and Tanner, 1997*). Pmt4 homodimers and Pmt1-Pmt2 heterodimers account for the major *O*-mannosylation activity on both soluble and membrane protein substrates (*Gentzsch and Tanner, 1997*; *Hutzler et al., 2007*). The loss of function of Pmt1 or Pmt2 is compensated for by the formation of alternative heterodimers, such as Pmt1-Pmt3 and Pmt5-Pmt2, which further explains the redundancy of PMT proteins in *S. cerevisiae* (*Girrbach and Strahl, 2003*).

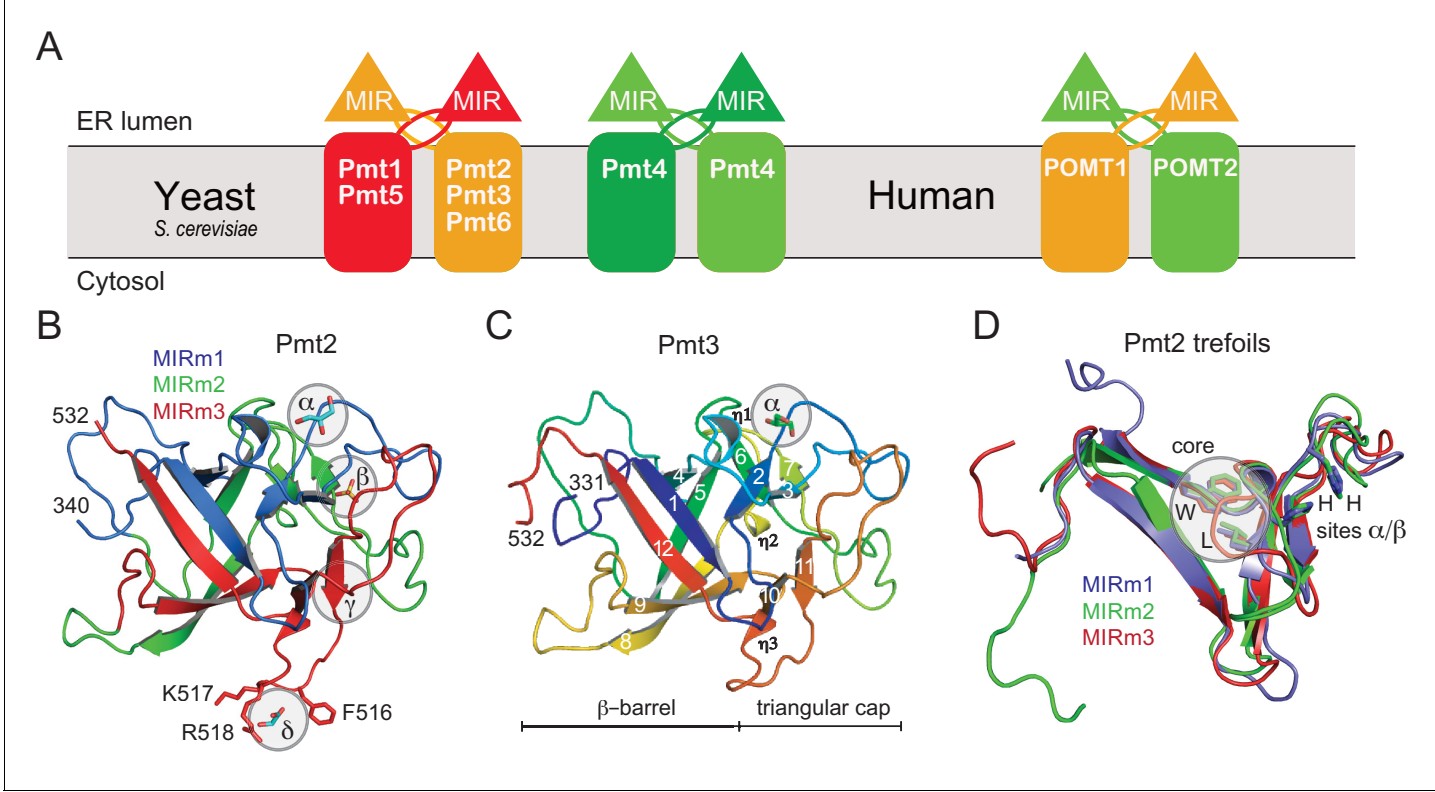

**Figure 1.** PMT-families and structure of PMT-MIR domains. (**A**) Schematic for the different PMT and POMT dimers. PMT1 (red), PMT2 (orange), and PMT4 (green) subfamilies are given together with the human homologs in the respective colors. In baker's yeast, PMT1 and PMT2 subfamily members form heterodimers, and Pmt4 homodimers. No partner is characterized for Pmt6. In mammals, the Pmt1 subfamily is not present. (**B**) Overall structure of Pmt2-MIR. Color coding is according to the MIR-motifs. Prominent ligand-binding sites ($\alpha$ to $\delta$) are highlighted. (**C**) The structure of Pmt3-MIR is almost identical to Pmt2-MIR. Color coding is in a ramp from N- (blue) to the C-terminus (red). Secondary structures are labeled and identical in Pmt2-MIR. (**D**) Superposition of the three MIR-motifs (MIRm1-m3) in the same color as in panel A. Each MIRm contributes a tryptophane and a leucine residue to the conserved core of the MIR domain. Sites $\alpha/\beta$, including two conserved histidine residues each, are only present in MIRm1 and MIRm2.

The online version of this article includes the following figure supplement(s) for figure 1:

**Figure supplement 1.** NanoDSF analyses of different Pmt-MIR domains.

**Figure supplement 2.** Comparison of MIR domains.

The majority of yeast proteins entering the secretory pathway are *O*-mannosylated (O-Man) by PMTs, including more than 90% of all cell wall proteins that usually carry numerous *O*-mannosyl glycans (*Neubert et al., 2016*). *O*-Mannosylation of PMT *bona fide* substrates occurs mainly in S/T-rich protein segments (*Larsen et al., 2017*; *Neubert et al., 2016*), whereas in ER protein quality control (referred to as unfolded protein *O*-mannosylation; UPOM) PMTs also target isolated serines and threonines of un- or misfolded proteins (reviewed in *Xu and Ng, 2015*). Remarkably, only Pmt1-Pmt2 is involved in UPOM (*Castells-Ballester et al., 2019*; *Xu and Ng, 2015*).

Despite the key roles of PMT-based *O*-mannosylation in multiple cellular processes, the molecular features driving Pmt1-Pmt2 based *O*-mannosylation of *bona fide* and UPOM target protein substrates have not been investigated. Only recently cryo-EM structures of the Pmt1-Pmt2 heterodimer from *S. cerevisiae* became available (*Bai et al., 2019*). In these structures each subunit has 11 transmembrane helices (TMHs) and two large hydrophilic loops oriented towards the ER lumen (LL): loop LL1 between TMH1 and 2 harboring a catalytic DE motif (*Lommel et al., 2011*), and loop LL4 between TMH7 and 8, encompassing a so-called MIR domain, whose function remains unknown.

MIR domains have been annotated originally from sequence homologies detected between mannosyltransferases, inositol triphosphate receptors (IP$_3$Rs) and ryanodine receptors (RyRs) (*Ponting, 2000*), which do not have a conserved protein function (*MacKrill, 1999*). MIR domains have a β-trefoil fold consisting of six β-hairpins arranged within a pseudo-threefold symmetry. This fold was

originally identified in interleukin-1β and fibroblast growth factors (*Murzin et al., 1992*) but it is also present in many carbohydrate-active enzymes (CAZy [*Lombard et al., 2014*]), such as the UDP-Gal-NAc:polypeptide GalNAc-transferases (GalNAc-Ts) that are responsible for mucin-type protein *O*-glycosylation in animals (*Gill et al., 2011*). Although not assigned yet, according to the CAZy classification (http://www.cazy.org), the MIR domain β-trefoils would belong to the carbohydrate-binding module (CBM) fold family 2 (CBM families 13 and 42; *Boraston et al., 2004*; *Fujimoto, 2013*; *Lombard et al., 2014*). Canonical CBM family 13 members are multivalent sugar binders and contain binding sites on each lateral face of the triangular domain named α, β, and γ (*Boraston et al., 2004*; *Fujimoto, 2013*). Structural and functional studies on PMT proteins carried out to date do not answer the question of whether PMT-MIR domains bind to the mannose of the Dol-P-Man substrate or the *O*-Man peptide product, and if or how they contribute to *O*-mannosylation of both *bona fide* substrates and UPOM target proteins.

Here, we combine structural biology and in silico analysis with in vitro and in vivo biochemistry to characterize the role of PMT-MIR domains in the *O*-mannosylation of both S/T-rich substrates and UPOM targets. Our data for the Pmt2-MIR domain and its comparison with other β-trefoils define the specifications of the PMT-MIR family, localize the binding site for mannosylated peptide products, and illustrate the role of unique PMT2 family MIR-motifs in the complementation of Pmt1-Pmt2 active sites. The importance of the MIR domain for *O*-mannosylation and disease are discussed in the context of the structure of the Pmt1-Pmt2 holoenzyme and related glycosyltransferases.

## Results

### PMT-MIR domain structures

The MIR domains of PMT1 and PMT2 family members (Pmt1, Pmt5, and Pmt2, Pmt3) were expressed in *Escherichia coli* and purified. While MIR domains of the PMT1 family proved to be unstable when analyzed by nano differential scanning fluorimetry (nanoDSF, *Figure 1—figure supplement 1*) and failed to crystallize, PMT2 family MIR domains were more thermostable and readily formed crystals. X-ray structures were determined by molecular replacement using the MIR domain of the stromal-derived factor 2 (SDF2) from *Arabidopsis thaliana* previously solved in our laboratory (*Radzimanowski et al., 2010*; *Schott et al., 2010*; *Figure 1B,C* and *Table 1*). The high resolution of the Pmt2-MIR (1.6 Å) and Pmt3-MIR (1.9 Å) structures allowed for a detailed analysis of the respective structural properties and ligand interactions. An X-ray structure of the Pmt2-MIR has been reported in a parallel study but was not analyzed for carbohydrate-binding and functional implications (*Bai et al., 2019*). The MIR domains of the PMT2 family reveal the canonical β-trefoil fold (12 β-strands in total, PFAM trefoil clan CL0066, *Figure 1C*; *Murzin et al., 1992*), comprising a six-stranded β-barrel and a triangular β-hairpin cap including a hairpin triplet. Despite a marginal sequence homology of below 20%, the PMT-MIR domains are structurally highly similar to the MIR domains of the IP$_3$R and RyR proteins (root mean square deviation (rmsd) of approx. 1.6 Å) (*Figure 1—figure supplement 2A*).

MIR domains consist of three trefoil MIR-motif units (*Bai et al., 2019*; *Schott et al., 2010*), in the following denoted MIRm1-3 (*Figure 1B,D*). Amino acid ranges for Pmt2-MIR are 339–401 (MIRm1), 402–466 (MIRm2) and 467–532 (MIRm3) (*Figure 2*). Each MIR-motif comprises two β-hairpins, one in the barrel and one in the cap, and an extended connection between the hairpins that completes the trefoil unit (*Figure 1D*). The hydrophobic core of the MIR domains involves one conserved tryptophan and leucine residue per MIR-motif that together constitute a central triad arrangement of the core at the barrel-cap interface. Given the high sequence and structural similarity between Pmt2- and Pmt3-MIR domains (68% identity, rmsd of 0.7 Å for 194 Cα-atoms), we focused further analysis on the Pmt2-MIR domain.

### The Pmt2-MIR domain contains three putative ligand-binding sites

In order to understand the role of PMT-MIR domains in protein *O*-mannosylation, we analyzed the Pmt2-MIR structure for specific surface properties. On the protein surface two regions reveal high sequence conservation (*Figure 3A*), in the following denoted as sites α and β (residue ranges see *Figure 2*) in accordance with general CBM terminology (*Boraston et al., 2004*). These sites localize to pronounced surface cavities in the cap region featuring a pair of histidines at their bottom. Both

**Table 1.** Data collection and refinement statistics for the Pmt-MIR domains.

| | Pmt2-MIR | Pmt2-MIR (low) | Pmt3-MIR |
|---|---|---|---|
| **Data collection** | | | |
| Space group | P 41 3 2 | I 21 21 21 | P 1 |
| Cell dimensions | | | |
| a, b, c (Å) | 139.6, 139.6, 139.6 | 35.4, 132.1, 132.7 | 55.3, 64.3, 65.6 |
| α, β, γ (°) | 90, 90, 90 | 90, 90, 90 | 107.9, 99.9, 99.7 |
| Resolution (Å) | 30.5–1.6 (1.66–1.6) | 93.6–2.3 (2.38–2.3) | 52.9–1.9 (1.97–1.9) |
| No. of total reflections | 579024 (54018) | 55295 (13703) | 61932 (5998) |
| Completeness (%) | 99.8 (100) | 95.9 (89.2) | 95.5 (92.3) |
| CC1/2 | 99.8 (52.7) | 96.8 (77.9) | 99.9 (82.4) |
| I/σ(I) | 13.45 (1.15) | 4.2 (1.4) | 14.14 (2.22) |
| $R_{pim}$ (%) | 3.8 (58.5) | 11.6 (43.3) | 3.1 (36.7) |
| Wilson B-factor (Å$^2$) | 20.8 | 25.2 | 27.7 |
| **Refinement** | | | |
| No. of reflections | 61502 (6045) | | 61926 (5997) |
| $R_{work}$ (%) | 14.1 (25.1) | | 17.4 (30.0) |
| $R_{free}$ (%)* | 15.7 (33.8) | | 22.0 (33.7) |
| No. of atoms | 1922 | | 6902 |
| Protein | 1591 | | 6516 |
| Ligands | 51 | | 18 |
| Solvent | 280 | | 368 |
| Protein residues | 194 | | 806 |
| B-factors (Å$^2$) | 31.6 | | 41.0 |
| Protein | 27.7 | | 40.8 |
| Ligands | 77.5 | | 49.1 |
| Solvent | 45.4 | | 43.8 |
| R.m.s. deviations | | | |
| Bond lengths (Å) | 0.014 | | 0.007 |
| Bond angles (°) | 1.16 | | 1.13 |
| Ramachandran (%) | | | |
| Favored | 95.8 | | 96.0 |
| Allowed | 4.2 | | 4.0 |
| outliers | 0 | | 0 |

*for 5% of all data.

cavities are rather negatively charged (*Figure 3B*). The histidines are part of conserved LH(S/T)H fin-fingerprints present in MIRm1 strand β2 and MIRm2 strand β6 (*Figure 2*). For Pmt2, H362 and H364 are fixed in a defined orientation by a complex hydrogen-bonding pattern within the strictly conserved site α in the interface between MIRm1 and MIRm3 (*Figure 3C*). MIRm1 site α, which is directly adjacent to the conserved L361 of the core, is completed by residues Y380, D384, N386, F504, and Q506 forming the wall of the cavity. The 384-DxNN sequence (x, any residue) is the second prominent conserved motif and constitutes the sole short helix (η1) within MIRm1 (*Figure 2*). An almost identical arrangement is found for Pmt2-MIR site β in the MIRm1-MIRm2 interface (*Figure 3D*). Both cavities are filled by ligands present in the crystallization solvent (glycerol in site α [also in Pmt3] and sulfate in site β).

Although Pmt2-MIR displays a pseudo-threefold symmetry typical for the β-trefoil fold classified as Type C CBMs (*Boraston et al., 2004*), the analogous site γ (the third binding site present in β-trefoils) is not present in MIRm3. Overall, MIRm3 differs the most in sequence, with mainly key-residues like L492 and W524 of the core being conserved (*Figure 2*). However, on the surface of the β11-β12 loop a two-residue insertion is present, which is specific to the PMT2 family (including Pmt2, Pmt3, and Pmt6) and human POMTs (*Figures 2* and *3E*). This loop forms a surface exposed ligand-binding site in Pmt2-MIR, from now on defined as PMT2 family-specific site δ. It consists of F516, K517, and R518, of which only the arginine is conserved in other PMT-families. The exposure of the

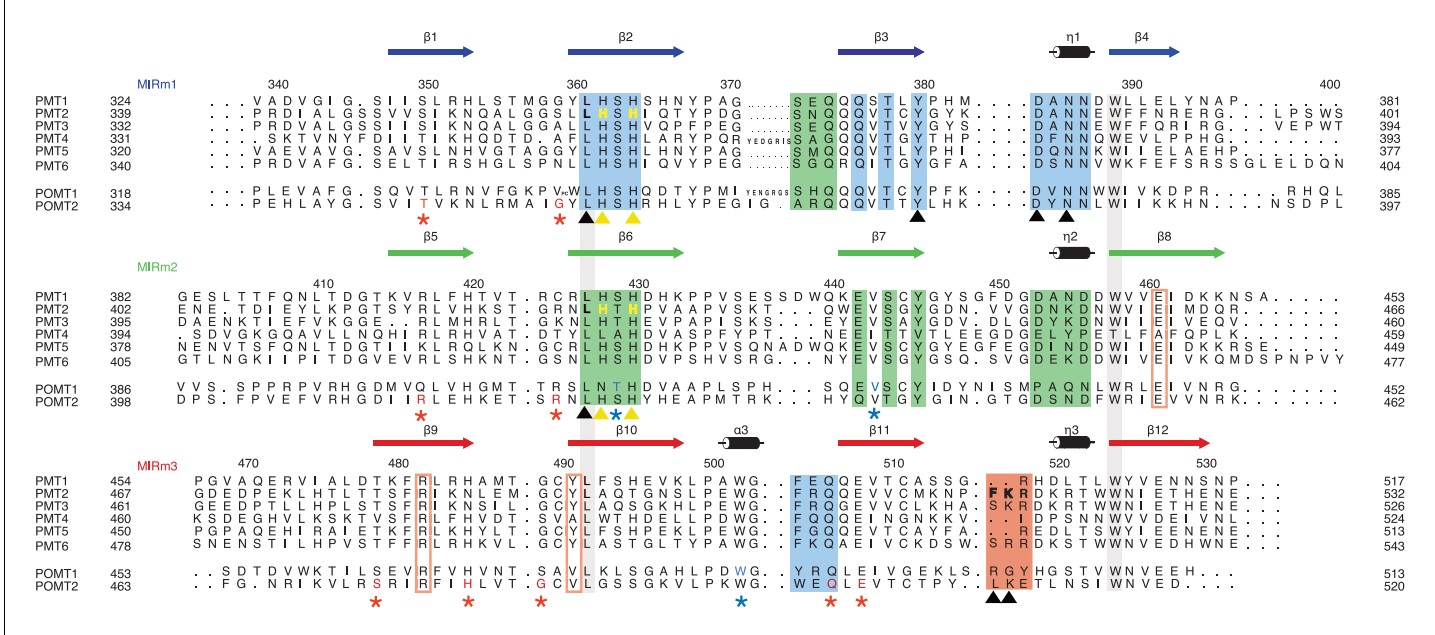

**Figure 2.** Sequence alignment of the PMT family. The alignment is structure-based for MIR domains of Pmt1, 2, and 3. Each line of the alignment corresponds to a MIR-motif and motifs are themselves aligned in order to highlight sequence conservations. Numbering and secondary structure above corresponds to Pmt2-MIR. The conserved leucine and tryptophane residues of the hydrophobic core in all MIR-motifs are highlighted by contiguous gray bars. Sites α, β, δ are highlighted by colored boxes and site δa by colored rectangles. Color code is according to *Figure 1B*. Residues mutated in this study are highlighted and marked with yellow and black triangles. Residues causing human pathologies are highlighted for POMT1 and POMT2 and marked with blue and red asterisks, respectively. All mutated residues are highlighted with colors matching the triangle or asterisk symbols.

hydrophobic phenylalanine is peculiar and indicative of a specific functional implication. The three residues form a trident that has picked another glycerol ligand from the solvent via hydrogen-bonding to the arginine side chain and to the backbone amides within site δ (*Figure 3E*). Overall, MIRm3 site δ displays a pronounced positive surface potential (*Figure 3B*). Adjacent to site δ, another three surface exposed residues are highly conserved (E461, R482, Y491; site δa) (*Figures 2* and *3A*), forming a tight patch that might act in concert with site δ. From superposition of Pmt2-MIR with MIR domains of RyR and IP$_3$R family members (*Figure 1—figure supplement 2B*), it is evident that sites α to δ are not conserved and the evolutionary correlation within the MIR family remains obscure.

Taken together, we identified three putative ligand-binding sites in the Pmt2- and Pmt3-MIR domains. Two MIR-motifs form almost identical surface cavities and are conserved in all PMT-MIRs, while the third site is specific for MIRm3 of the PMT2 family and is highly solvent exposed.

## The Pmt2-MIR domain represents a putative CBM

Although MIR domains comprise the β-trefoil fold that has been generally annotated as CBM fold family 2 (*Boraston et al., 2004*), binding of any carbohydrate to MIR domains has not yet been reported. In order to characterize ligand binding of the Pmt2-MIR domain in more detail, we performed a structure-based homology search to find related carbohydrate binders. Besides the relationship of the PMT-MIRs to SDF2 (*Schott et al., 2010*) and within the MIR family, we found a close structural homology to many lectins with small-sugar-binding Type C CBMs (*Boraston et al., 2004*). Other structurally related carbohydrate-binding β-trefoils were found in xylanases (PDB ID: 1isv) or galactosidases (PDB ID: 1ups), thus justifying the general β-trefoil classification as CBMs in the CAZy database (*Lombard et al., 2014*).

In all compared β-trefoil structures, carbohydrate binding almost exclusively occurs at the triangular cap region (the hairpin triplet) and not at the β-barrel (*Figure 4A*). Most importantly, the analysis shows that the binding modes to sites α and β are highly conserved. However, a corresponding carbohydrate-binding γ site within the third trefoil (MIRm3) is not present in Pmt2-MIR and is generally

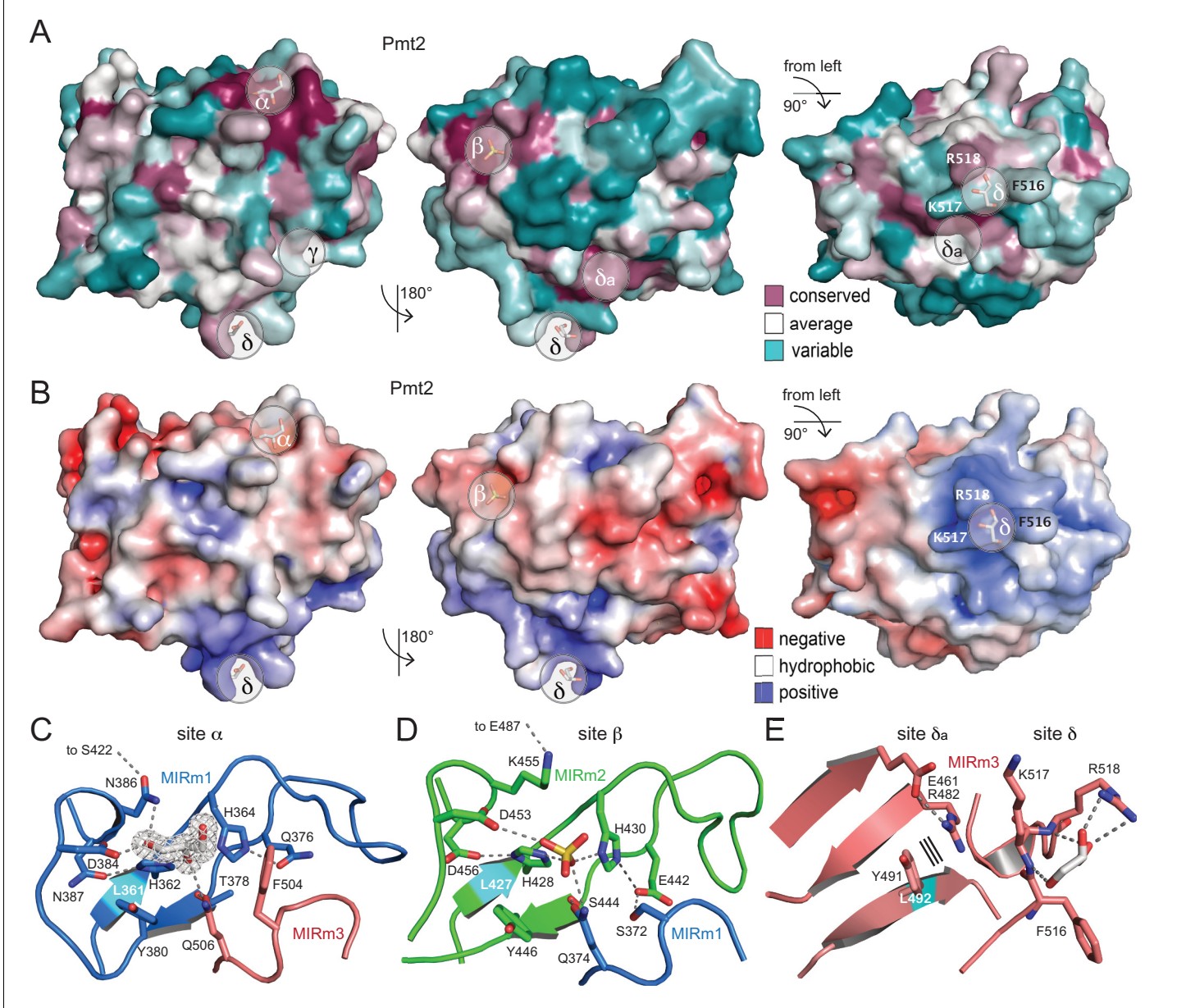

**Figure 3.** Surface properties and ligand-binding sites of Pmt2-MIR. (**A**) Surface conservation within the PMT family (increasing conservation from cyan over white to purple). Sites α, β, and δa correspond to regions of highest conservation, while sites γ and δ are PMT2-specific. (**B**) Same views for the electrostatic surface potential of Pmt2-MIR (±5 $_kBT$). While sites α and β are rather negatively charged (red), site δ forms a positively charged patch (blue). (**C**) The strictly conserved site α of the PMT family in the MIRm1/MIRm3 interface. Two highly coordinated histidines at the bottom of the cavity of the site are ligated to a glycerol ligand, which is shown within its $2mF_o\text{-}DF_c$ electron density (2σ). Hydrogen bonds are shown by dashed lines. The conserved leucine of the core is highlighted in cyan. (**D**) The almost identical site β in the MIRm2/MIRm1 interface harbors a sulfate ion. (**E**) Pmt2-MIR-specific site δ and PMT-conserved site δa within MIRm3. Site δ forms a highly exposed trident (F516, K517, R518) that coordinates a glycerol molecule, while the adjacent site δa is ligand free. Parallel lines indicate stacking.

not conserved within the PMT family members (*Figure 2*). Ligands found in Pmt2-MIR sites α and β match the sugar moieties in the compared structures (*Figure 4B and C*). The general theme, as typical for CBM-carbohydrate interactions (*Boraston et al., 2004*), is the packing of a sugar ring (glycerol in Pmt2-MIR) against a solvent exposed aromatic residue (Y380 in Pmt2-MIRm1), while the hydroxyl-groups are bonded to a polar residue at the bottom of the sugar-binding cavity (H362 in Pmt2-MIRm1).

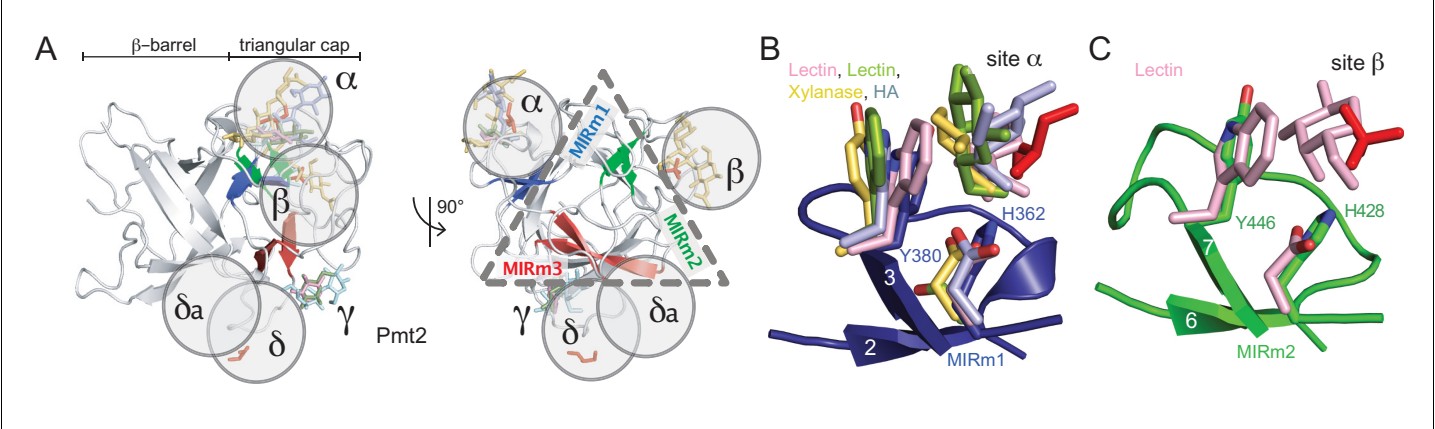

**Figure 4.** The MIR domain as carbohydrate-binding module (CBM). (**A**) The Pmt2-MIR domain is shown with only the β-strands within the triangular cap colored according the respective MIRm. Carbohydrate moieties bound to the α, β, and γ sites in β-trefoil CBM structures are superposed and shown together with the ligands bound to Pmt2-MIR sites α to δ (red). Each ligand-binding site locates to one lateral surface of the triangle. (**B**) Site α corresponds to a general carbohydrate-binding site and presents a conserved aromatic and a polar residue. (**C**) Although very similar to site α, site β is less frequently occupied by carbohydrate ligands.

In summary, structure-based homology demonstrates the feasibility of ligand binding in Pmt2-MIR to the general β-trefoil carbohydrate-binding sites α and β, suggesting its interaction with the mannose moiety of donor or acceptor substrates.

## The Pmt2-MIR domain interacts with O-Man peptide ligands

To characterize the ligand-binding capacities and specificities of Pmt2-MIR, we performed NMR interaction studies with both substrates and products of the enzymatic reaction. Isotopically labeled ($^{15}$N and $^{13}$C,$^{15}$N) Pmt2-MIR was produced. By using 3D triple-resonance experiments in combination with selective $^{15}$N-labeling of amino acids (*Figure 5—figure supplement 1*), more than half (56%) of the backbone amide resonances could be assigned to structurally map putative interactions with mannose and derivatives by NMR screening and titration experiments. The chemical shift perturbations (CSPs) show weak, but specific binding of mannose and α-mannose-1-phosphate (*Figure 5—figure supplement 2*). None of the other sugars tested (glucose, glucose-1-phosphate, glucose-6-phosphate, methyl-α-glucoside, mannose-6-phosphate and methyl-β-mannoside) showed significant CSPs even at a concentration of 10 mM.

The determined interaction surface encompassed two spatially close regions belonging to Pmt2-MIRm1 and -MIRm2 (*Figure 5—figure supplement 3*). The largest CSPs upon interaction with mannose are observed in and around the Pmt2-MIRm2 453-DNKD motif (D453 and D456 show the largest CSPs), which correspond to site β. Small but significant CSPs are also observed in Pmt2-MIRm1 (S359, L360, L361, R395, and G396) and its close surroundings (S422, T495, and F504) indicating the additional involvement of site α in the interaction.

Following these results, NMR titration experiments were performed to monitor whether Pmt2-MIR binds O-Man acceptor peptides (*Bausewein et al., 2016*; *Weston et al., 1993*). No significant changes in the $^{1}$H,$^{15}$N-correlation spectra were observed upon addition of eight times excess of the unmodified peptides (YATAV and CYATAV) indicating that these do not interact (*Figure 5B*). In contrast, the clear appearance and disappearance of signals and CSPs upon addition of the O-Man peptides (YAT(O-Man)AV and CYAT(O-Man)AV), which are already visible at a 1:1 Pmt2-MIR:peptide ratio, indicate an interaction on the intermediate to fast exchange NMR timescale. The affinity of Pmt2 for the O-Man peptides is much stronger than for the mannose alone. Mapping of the O-Man peptide binding on the Pmt2-MIR crystal structure shows that the same two regions are involved as for mannose, albeit to a different extent. The backbone amides showing the largest CSPs (slow exchange) are observed around site α (*Figure 5A,B*), which suggests this is the main region recognizing the O-Man peptide. Interestingly, upon interaction with mannose as well as with the O-Man peptide, the backbone amides of the two highly conserved leucine residues in the MIRm1 and

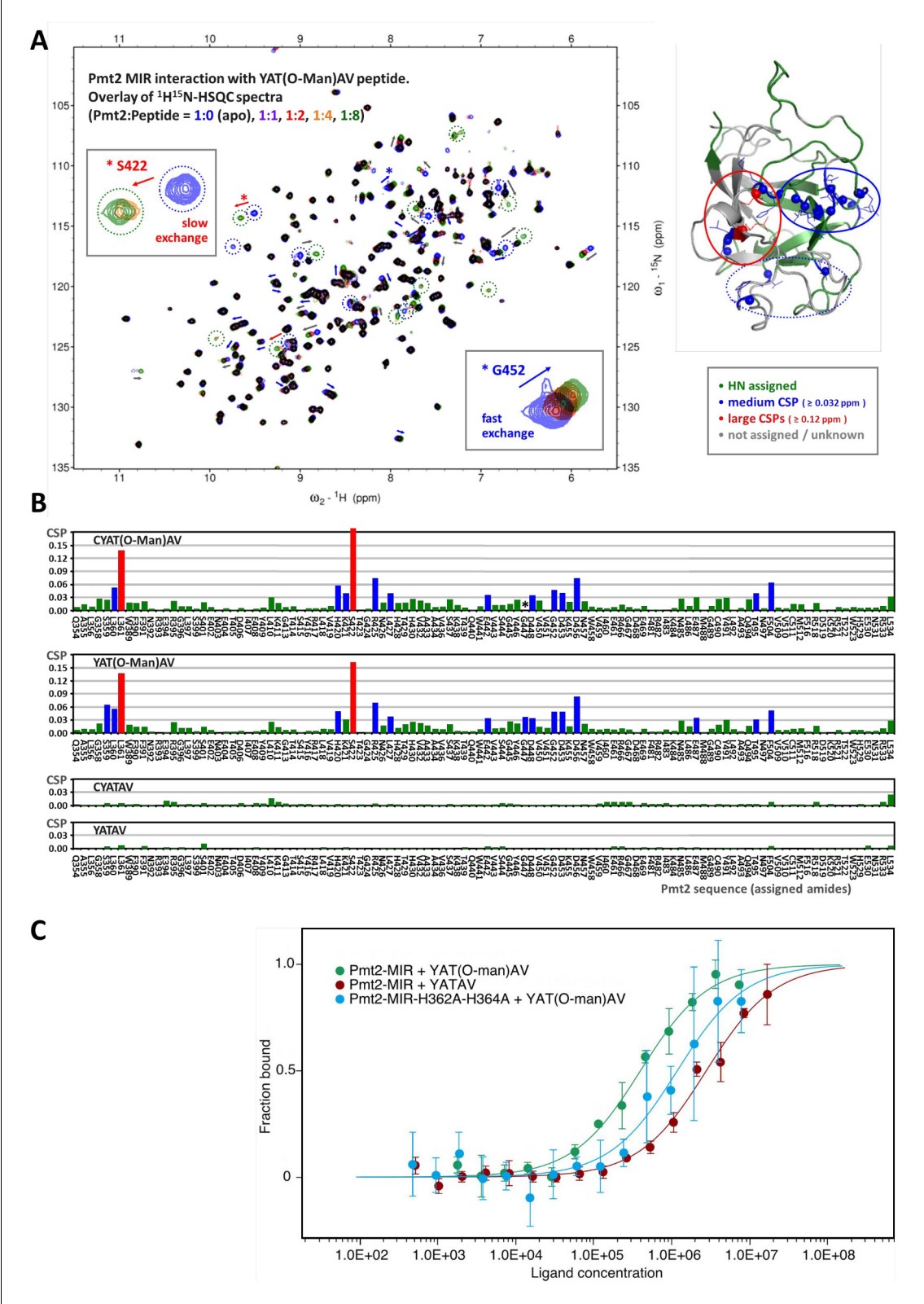

**Figure 5.** Characterization of O-Man peptide binding to Pmt2-MIR. (**A, B**) NMR analyses. (**A**) Interaction of Pmt2-MIR with the YAT(O-Man)AV peptide (NMR titration experiment using ¹H¹⁵N-HSQC spectra, left). Disappearing and new appearing signals and evident chemical shift perturbations (CSPs) indicate binding of the O-Man peptide. The most perturbed amide cross peaks are mapped onto the X-ray crystal structure (right) showing that mainly two (conserved) regions from MIRm1 and MIRm2 are involved (corresponding to sites α (in continuing red circle) and β (in continuing blue circle) albeit

*Figure 5 continued on next page*

*Figure 5 continued*

to a different extent. Interestingly, the backbone amides of some conserved leucine residues buried within the interior of the protein are also affected upon the interaction, hinting at indirect effects such as conformational changes within the protein upon substrate binding. (B) Pmt2-MIR interaction with O-Man and unmodified peptides. The CSPs for the Pmt2-MIR interaction with O-Man and unmodified peptides were obtained from NMR titration series ($^1$H$^{15}$N-HSQC spectra measured at Pmt2-MIR:peptide ratios of 1:0 (apo), 1:1, 1:2, 1:4, and 1:8). The combined $^1$H$^{15}$N amide CSPs ($\Delta\delta HN = \sqrt{[(\Delta\delta H)^2 + (0.1\times\Delta\delta N)^2]}$, in $^1$H ppm) are mapped on the sequence of Pmt2 showing no significant changes for the unmodified peptides and evident CSPs for the O-Man peptides (YAT(O-Man)AV and CYAT(O-Man)AV). An overall decrease of the amide signal intensity is observed for the Pmt2-MIR titration with the CYAT(O-Man)AV peptide (G447, marked with an asterisk, cannot be traced in the spectra as it disappears). (C) MST analysis of peptide substrate/product binding to Pmt2-MIR. The YATAV peptide and its O-Man product YAT(O-Man)AV are analyzed for Pmt2-MIR and its site α mutant lacking both conserved histidines (H$_{362}$A, H$_{364}$A).

The online version of this article includes the following figure supplement(s) for figure 5:

**Figure supplement 1.** Selective $^{15}$N-labeling of amino acids in Pmt2-MIR.
**Figure supplement 2.** NMR screening of Pmt2-MIR sugar interactions.
**Figure supplement 3.** Pmt2-MIR interaction with mannose.
**Figure supplement 4.** Interaction of Pmt2-MIR (A) and Pmt5-MIR (B) with the CYAT(O-Man)AV peptide.

MIRm2 (L361 and L427, respectively) domain show evident CSPs, whereas the leucine in MIRm3 (L492) does not show any significant perturbation. Even though the leucine side chains reside in the core of the protein, the backbone amide senses (indirect) effects upon the interaction with mannose and the O-Man peptides.

Based on these results, the binding of the O-Man peptide CYAT(O-Man)AV to the MIR domain of the PMT1 family member Pmt5 was compared to Pmt2-MIR by NMR titration using 1D $^1$H spectra. The two MIR domains show different interaction profiles to the O-Man peptide, which are indicative for distinct NMR timescales. While Pmt2-MIR strongly binds on an intermediate timescale (signals broaden and disappear; *Figure 5—figure supplement 4A*), Pmt5-MIR binds in slow-exchange (disappearing and newly appearing signals for the free and bound form respectively; *Figure 5—figure supplement 4B*). These data suggest that the MIR domains of distinct families exhibit a different affinity or specificity and possibly operate at different kinetic timescales.

Taken together, the NMR data pinpoint that two adjacent regions within Pmt2-MIR (corresponding to sites α and β) are distinctively involved in the interaction with mannose and O-Man peptides. Moreover, the Pmt5-MIR also binds the O-Man peptide, though on a different NMR timescale, which may be relevant for the specificity, activity, and interplay of the MIR domains in protein *O*-mannosylation.

## Pmt2-MIR site α prefers mannosylated peptides

To test whether Pmt2-MIR site α is the binding site for the mannosylated peptide product and to quantify binding affinities, we performed microscale thermophoresis (MST) measurements (*Seidel et al., 2013*). This method allowed us to determine dissociation constants ($K_D$ values) for fluorescently labeled Pmt2-MIR and YATAV or YAT(O-Man)AV peptides in order to compare affinities for the substrate or product of the mannosylation reaction (*Figure 5C*). The fluorescence signal enabled measurements of labeled Pmt2-MIR at a typical concentration of 100 nM and a ligand concentration ranging from high nanomolar to low millimolar concentrations. For the unmodified peptide, MST measurements yield a weak binding affinity in the low millimolar range ($K_D$ of 2.8 ± 0.4 mM), whereas the O-Man peptide binds about seven times stronger (400 ± 55 μM), demonstrating that Pmt2-MIR is involved in product binding rather than in substrate recognition. To narrow down the interaction site and to challenge site α in sugar recognition, the two conserved histidines (H362, H364) at the bottom of the putative sugar-binding pocket were mutated to alanines. Performing the MST analysis with an identical parameter setup shows a threefold decrease in affinity for the O-Man peptide (1.2 ± 0.3 mM) (*Figure 5C*). The preferred binding over the unmodified peptide can be explained by the remaining wall of the cavity (e.g. Y380) at site α as described above. Interestingly, although the overall protein abundance of Pmt2 carrying the histidine double mutation is not apparently affected in vivo (*Figure 7—figure supplement 1A*), nanoDSF measurements for recombinant Pmt2-MIR2 (*Figure 1—figure supplement 1*) suggest that the mutation also causes domain destabilization.

Overall, the peptide binding data are in agreement with the prediction from comparative structural analysis and the NMR data, and suggest that Pmt2-MIR site α represents the preferred binding site for O-Man peptide products.

### Analogy between Pmt2-MIRm1 site α and the GalNAc-T2 lectin domain

GalNAc-Ts are a prime example of the dynamic interplay between catalytic and lectin domains necessary for efficient protein O-glycosylation (*Lira-Navarrete et al., 2015*). The general function of the lectin-like β-trefoils is thought to be the modulation of local substrate concentration for efficient catalysis and orchestration of S/T-rich multi-acceptor substrates (*Dupont et al., 1998*; *Fujimoto et al., 2002*; *Thobhani et al., 2003*). The structure of GalNAc-T2 (*Lira-Navarrete et al., 2015*; *Lira-Navarrete et al., 2014*) in complex with a glycopeptide reveals two GalNAc moieties, one in the active center and one in site α of the lectin domain, bound to two threonine residues spaced by nine residues (*Figure 6A*). In accordance with our structural analyses, the binding mode of the glycopeptide to the lectin domain might serve as a first model for the recognition of a O-Man peptide by Pmt2-MIR site α (*Figure 6B*). Moreover, when we compare the GalNAc-T2 structure with the Pmt1-Pmt2 cryo-EM reconstruction (*Bai et al., 2019*), the distances between the active centers and the β-trefoil sites α are suitable to accommodate glycopeptides with sugars in similar spacing (*Figure 6C*).

Taken together, the analogy of the PMT-MIR domain to the GalNAc-T2 lectin domain suggests similar binding modes and function of the β-trefoil in modulating local substrate concentration for efficient catalysis of O-glycosylation.

### Probing the function of Pmt2-MIR ligand-binding sites in vivo

We further tested the functional importance of Pmt2-MIR ligand-binding sites for Pmt2 activity in vivo by alanine substitutions of histidine residues in the conserved LH(S/T)H fingerprints to which carbohydrate binding was assigned in MIRm1 site α (H362, H364) and MIRm2 site β (H428, H430), as well as of the PMT2-subfamily-specific MIRm3 FKR motif in site δ (F516, K517). All Pmt2 mutant proteins are equally abundant in total membranes when compared to wild type Pmt2, indicating that the stability of full length Pmt2 is not affected by the respective amino acid substitution (*Figure 7— figure supplement 1A*). To assess the impact of the Pmt2-MIR mutations on mannosyltransferase activity of the Pmt1-Pmt2 complex in vivo, the O-mannosylation status of canonical S/T-rich Pmt1-

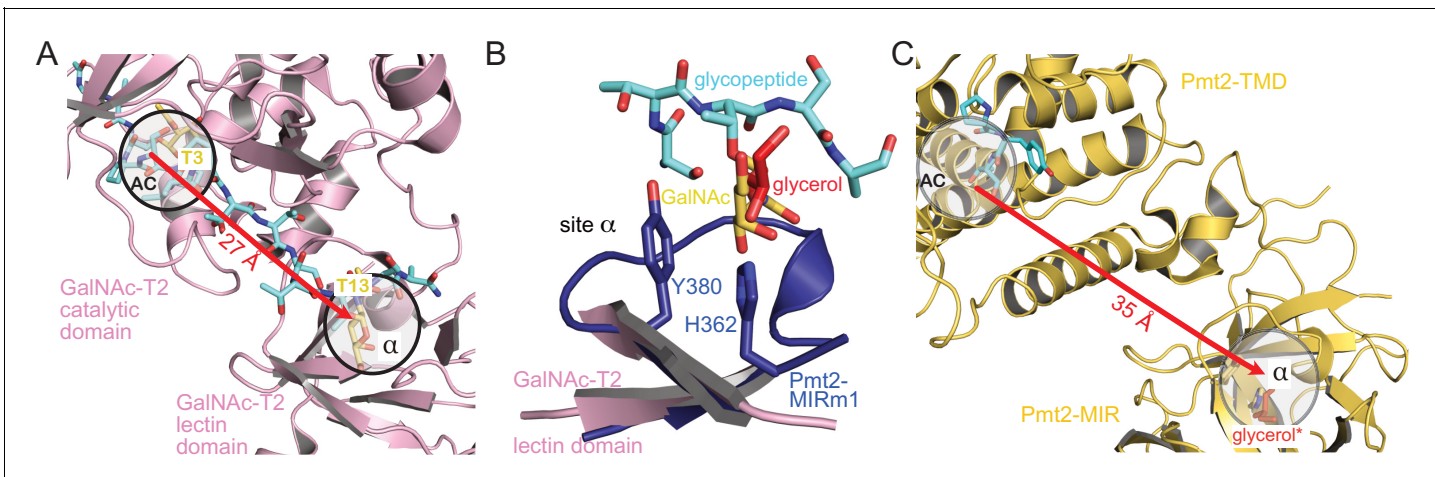

**Figure 6.** Analogy of β-trefoils in Pmt2 and GalNAc-T2. (A) Spatial correlation of the active center (AC) and site α within the lectin domain of human GalNAc-transferase 2 (pink) bound to a GalNAc-glycopeptide (PDB-ID 5ajo [*Lira-Navarrete et al., 2015*]). Both sites are occupied by GalNAc moieties (yellow) attached to threonine T3 and T13 of the glycopeptide (cyan), respectively. The site distance is indicated by a red arrow. (B) Superposition of Pmt2-MIRm1 site α (blue) with the lectin domain of human GalNAc-T2. The glycerol (red) within Pmt2-MIRm1 site α overlaps with the GalNAc moiety of the glycopeptide. (C) Spatial arrangement of the AC within the Pmt2-TMD in respect to site α of Pmt2-MIR in the Pmt1-Pmt2 complex (PDB-ID 6p25 [*Bai et al., 2019*]). Pmt2-TMD is bound to the PYTV-peptide, and Pmt2-MIR site α is shown occupied with glycerol as found in the Pmt2-MIR X-ray structure (red, superposition denoted by an asterisk). The site distance is similar to GalNAc-T2.

Pmt2 substrates (the extracellular heat shock protein Hsp150 [*Russo et al., 1993*] and the cell wall protein Scw4 [*Cappellaro et al., 1998*]) was analyzed by Western blotting. Decreased *O*-mannosylation in the absence of Pmt2 is indicated by a shift of the target protein to lower molecular masses as previously described (*Gentzsch and Tanner, 1997*; *Grbavac et al., 2017*). Among the mutants tested, only alanine substitutions of histidines in MIRm1 site α (H$_{362}$A, H$_{364}$A [Pmt2-2His-α]) affected *O*-mannosylation of Hsp150 and Scw4 (*Figure 7A*).

In addition to *O*-mannosylation of S/T-rich substrates, the Pmt1-Pmt2 complex is also known to act on un- or misfolded proteins which are normally not targets of *O*-mannosylation, a process that is referred to as UPOM (reviewed in *Neubert and Strahl, 2016*). To investigate the effect of the histidine mutations in PMT2-MIR sites α and β on UPOM, we took advantage of ER-GFP, which undergoes *O*-mannosylation by Pmt1-Pmt2 in the ER due to its slow folding kinetics. *O*-mannosyl glycans disturb proper folding of the fluorophore resulting in decreased fluorescence intensity and rendering this protein an excellent reporter to monitor UPOM efficiency (*Castells-Ballester et al., 2019*; *Xu et al., 2013*). Intriguingly, respective histidine substitutions (Pmt2-4His-α/β) do not affect *O*-mannosylation of ER-GFP as indicated by Western blot (*Figure 7B*, lower panel and *Figure 7—figure supplement 1B*) and by the highly similar fluorescence intensities of ER-GFP (*Figure 7B*, upper panel) in strains expressing wild type and mutated Pmt2 versions.

Alanine mutants of the PMT2 family-specific FKR motif (FK- δ: F$_{516}$A, K$_{517}$A) do not influence *O*-mannosylation of the canonical Pmt1-Pmt2 target proteins (Hsp150 and Scw4; *Figure 7A*) and of the UPOM target ER-GFP (*Figure 7B*). Nevertheless, the K$_{517}$A mutant protein does not fully complement the temperature sensitivity of mutant *pmt2pmt4* (*Figure 7C*) suggesting a minor, but significant impact on Pmt2 function. Similar phenotypes were observed for conserved leucine residues in the PMT2-MIR sites α and β and the 384-DxNN motif (*Figure 7—figure supplement 1C*).

In summary, our in vivo data show that the histidines in PMT2-MIRm1 site α are particularly important for *O*-mannosylation of S/T-rich substrates, but not for non-canonical protein substrates during ER protein quality control.

## Discussion

In the present study, we derive a general working model for the regulation of *O*-mannosylation by PMT-MIR domains in the context of the full length Pmt1-Pmt2 protein complex. Recent cryo-EM structures of the yeast Pmt1-Pmt2 heterodimer (*Bai et al., 2019*) did not assign a function to the MIR domains, leaving the open question of if or how they contribute to regulation of the PMT catalytic mechanism. Our analysis of high-resolution X-ray structures of the Pmt2- and Pmt3-MIR domains revealed particular ligand-binding sites (α, β, δ) on each lateral side of the triangular cap region. Generally, although the β-trefoil fold is a dedicated CBM (*Boraston et al., 2004*; *Fujimoto, 2013*), carbohydrate-binding by PMT-MIRs had not been investigated. Sequence and structural conservation within sites α and β in other PMT-MIR domains and carbohydrate-binding β-trefoils prompted us to conduct an in-depth investigation into their binding propensities for mannose derivates and *O*-Man peptides. NMR investigations revealed that Pmt2-MIR is indeed a CBM and two adjacent regions are distinctively involved in the specific interaction with mannose, α-mannose-1-phosphate and *O*-Man peptides. While mannose weakly interacts preferentially with MIRm2 site β, the *O*-Man peptides show a much stronger binding particularly to MIRm1 around site α. Furthermore, the divergent interaction of Pmt5-MIR of the PMT1 family with the *O*-Man peptide may be related to the dynamic interplay and different specificity of the PMT families.

The complete Pmt1-Pmt2 cryo-EM structure (*Bai et al., 2019*) revealed an asymmetric MIR domain arrangement with Pmt1-MIR touching the Pmt2-transmembrane domain (TMD) and Pmt2-MIR not forming any tertiary contact (*Figure 8A*). However, the architecture of the α and β sites is conserved between Pmt1 and Pmt2 suggesting functional accordance. Our analysis for Pmt2-MIR shows that both sites are able to bind carbohydrates, while the respective site γ in MIRm3 is sterically blocked. In the cryo-EM structure, the Pmt1-MIR γ site constitutes a major part of the interface with the PMT2-TMD next to the active center (*Figure 8—figure supplement 1A*) and the interface is completed by a PMT1-MIR domain-specific loop insertion (Pmt1 421-SDW) (*Figure 2*). Strikingly, on the opposing side this contact involves a PMT2 family-specific four residue insertion within loop LL4 (Pmt2 570-PDKF) that links the MIR domain to the TMD (*Figure 8—figure supplement 1B*). This insertion is complementary to Pmt2-MIRm3 site δ, exposing an aromatic residue conserved in the

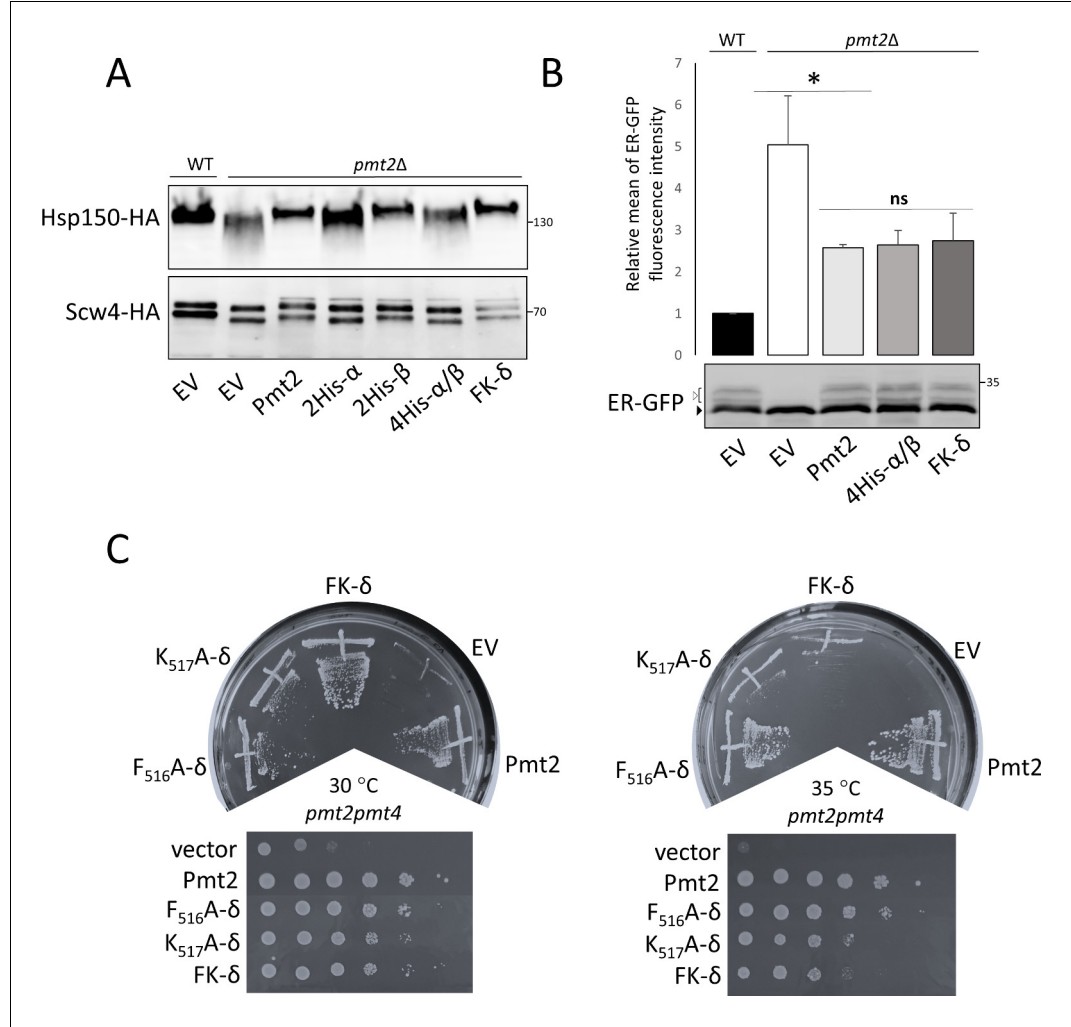

**Figure 7.** Functional characterization of Pmt2-MIR in vivo. (**A-C**) Analysis of yeast strains expressing a vector control (YEp352; empty vector; EV) or HA-epitope tagged versions of wild type Pmt2 (pVG80) or mutant thereof with alanine substitutions in site α (2His-α: $H_{362}A$, $H_{364}A$ (pAG1)), site β (2His-β: $H_{428}A$, $H_{430}A$ (pAG2)) and in combination (4His-α/β: $H_{362}A$, $H_{364}A$, $H_{428}A$, $H_{430}A$ (pAG4)), as well as in site δ ($F_{516}A$ (pAG7); $K_{517}A$ (pAG8); FK-δ: $F_{516}A$, $K_{517}A$ (pAG9)). (**A**) *O*-mannosylation status of canonical Pmt1-Pmt2 substrates Hsp150 (upper panel) and Scw4 (lower panel). Indicated constructs were expressed in *pmt2Δ* mutants expressing HA-tagged Hsp150 (strain AGY15, upper panel) or Scw4 (strain *pmt2Δ* transformed with YEp351a, lower panel). Hsp150 and Scw4 were isolated and analyzed by Western blot as detailed in Material and methods. Molecular weight marker is indicated on the right. (**B**) Analysis of the *O*-mannosylation status of the UPOM substrate ER-GFP by FACS (upper panel) and Western blot (lower panel). Indicated constructs were expressed in a wild type strain (BY4741; WT) or a *pmt2Δ* mutant (Y00386). Upper panel: Fluorescence intensities of the strains are shown, indicating the degree of *O*-mannosylation of ER-GFP. The mean fluorescence intensity value of WT strain was taken as reference and set to one. Statistical analysis was performed using t-test at the significance level p<0,05 (*). Abbreviation *ns* stands for not significant. Lower panel: Different glycosylated variants of ER-GFP are visible (black and white arrows) that have been described previously (*Castells-Ballester et al., 2019*). (**C**) Complementation of thermosensitive phenotype of *pmt2pmt4* mutant transformed with the indicated constructs. Thermosensitivity of the strains was assessed 45 hr after the inoculation by comparing yeast grown at 30˚C and 35˚C. Tenfold serial dilutions, starting from $10^6$, are shown in lower panels.

The online version of this article includes the following figure supplement(s) for figure 7:

**Figure supplement 1.** Characterization of Pmt2 alanine-exchange mutants.

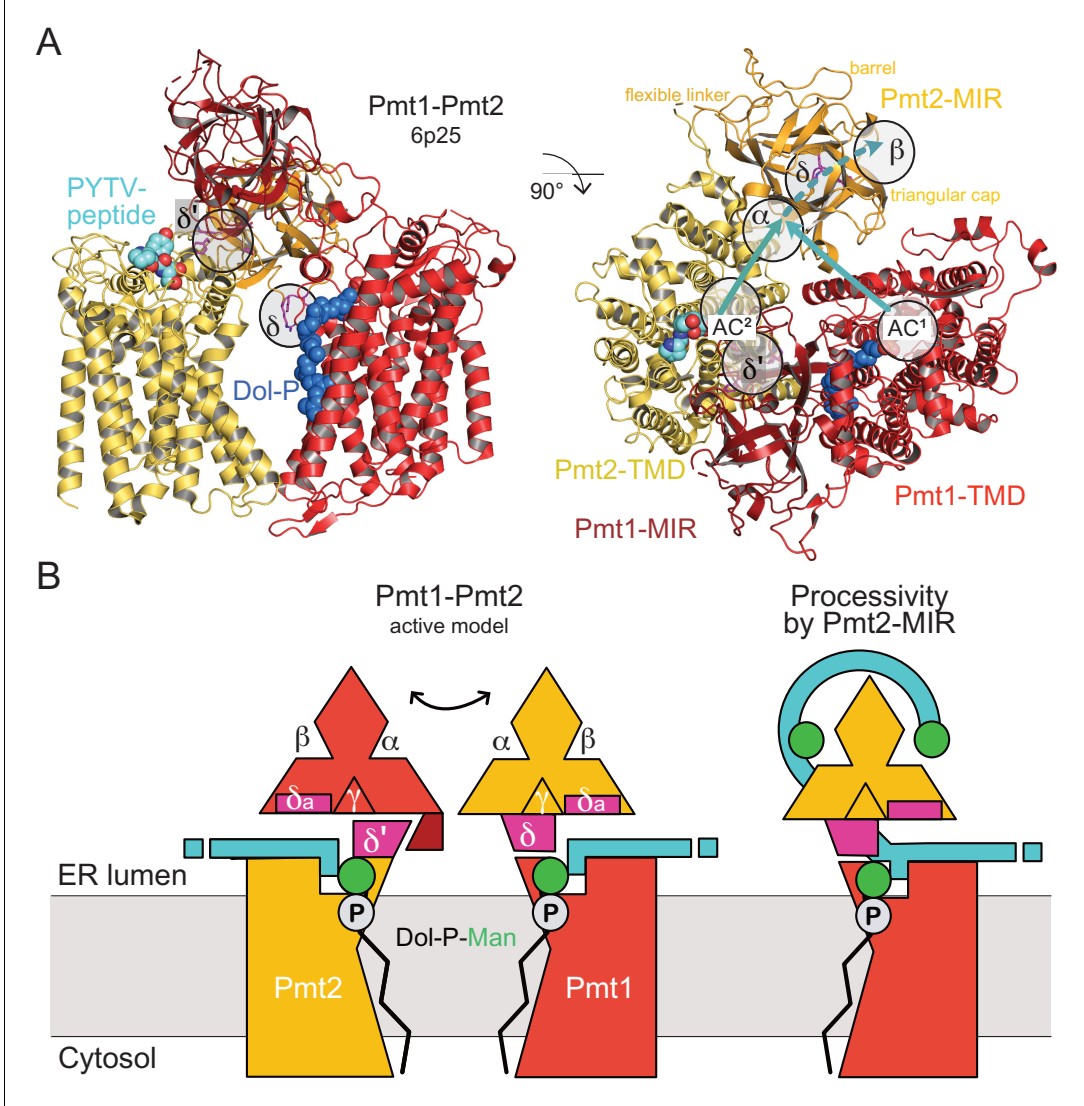

**Figure 8.** MIR domain function in protein *O*-mannosylation. (**A**) Cryo-EM structure of the yeast Pmt1-Pmt2 complex (PDB-ID 6P25 [*Bai et al., 2019*]). Substrate-peptide fragment and dolichol-phosphate (Dol-P) are bound to different subunits and MIR domain orientation is asymmetric. ligand-binding sites and the location of the active sites of Pmt1 and Pmt2 (AC$^1$, AC$^2$) are indicated. Putative trajectories of mannosylated peptides from both ACs to site α and subsequently site β within Pmt2 are indicated by cyan arrows. Site δ (magenta, FKR shown in sticks) might functionally correspond to site δ' of Pmt2-TMD involved in Pmt1-MIR interactions. (**B**) Schematic models for active Pmt1-Pmt2 dimer and for Pmt2-MIR processivity. Left: Sites δ and complementary δ' mediate alternate TMD-MIR domain interactions (domain dynamics indicated by double-arrow). Sites δa might interact with the substrate (cyan). Right: Sites α and β of Pmt2-MIR bind to the mannosylated peptide thus organizing the active center of Pmt1 for processive reaction. The online version of this article includes the following figure supplement(s) for figure 8:

**Figure supplement 1.** Domain interactions in the Pmt1-Pmt2 heterodimer.
**Figure supplement 2.** Missense mutation sites in human POMT1/2.

PMT2-family (Pmt2 F573), and we thus denote it as site δ'. When we superimpose the Pmt2-MIR structure on Pmt1-MIR in the cryo-EM structure and also swap the TMDs, Pmt2-MIR touches the Pmt1-TMD with a corresponding interface (*Figure 8—figure supplement 1C*). In doing so, site δ spatially corresponds to the LL4 insertion and the two phenylalanines match exactly. Finally, the high conservation of site δa (next to site δ) within all PMT-MIR domains still remains enigmatic, but due to its position in close proximity to the active site it might be involved in substrate-peptide recognition. The overall Pmt1-Pmt2 geometry including the new features identified in this study, is schematized

in *Figure 8B*. Taken together, dynamics within the PMT-specific interfaces of the MIRm3 trefoils with the MIR-TMD linker regions next to the active centers might contribute to catalysis.

The dynamic interplay between catalytic domains and carbohydrate-binding β-trefoils is not a novel concept for carbohydrate-active enzymes and has been demonstrated for lectin-like β-trefoils in GalNAc-Ts (*Lira-Navarrete et al., 2015*; *Lira-Navarrete et al., 2014*). Following the idea of dynamic interactions, our data for Pmt2-MIR provide also a first rationale for the specific function of sites α and β conserved in all PMT-MIR domains. When we analyze our data in the context of the Pmt1-Pmt2 cryo-EM structure (*Bai et al., 2019*), the PMT2-MIR site α is equidistant to site β and both active centers (*Figure 8A*), well suited to bind the reaction product with modified serine residues spaced about ten residues apart. Whether the peptide is mannosylated by the active center in Pmt1 or Pmt2 remains elusive as the reaction mechanism of the PMT dimers is not resolved and the substrate trajectories cannot be inferred from static superimpositions.

In vivo data presented here complement the structural interpretation and provide first insights into the molecular determinants driving PMT enzymology. The in vivo Pmt2 mutational analyses demonstrate that site α is relevant only for the *O*-mannosylation of S/T-rich substrates, however it seems to be dispensable for the *O*-mannosylation of unfolded protein targets. Altogether these data establish that PMTs can act as processive enzymes in which the highly conserved site α is key to catalysis most likely by binding the O-Man products to ensure efficient mannosylation of S/T-rich substrate domains (schematized in *Figure 8B*). Whether the function of the MIR domain is to keep the reaction product away from the active center and/or to increase the local concentration of nearby unmodified substrate serine and threonine residues remains to be seen.

Recently, a new class of protein O-mannosyltransferases, encoded by the *TMTC1-4* genes, has been identified in metazoan, which specifically mannosylate distinct serine and threonine residues in the β-strands within extracellular domains of cadherin superfamily members (*Larsen et al., 2017*). TMTCs (GT family 105) are multispanning transmembrane ER proteins with a variable number of C-terminal tetratricopeptide repeats (TPRs) that are assumed to be important for interactions between the mannosyltransferase and its protein substrates (*Larsen et al., 2019*). These enzymes resemble PMTs in number and topology of TMDs, as well as amino acid conservation in certain loop regions, although they lack MIR domains (*Albuquerque-Wendt et al., 2019*). This might explain the different substrate specificities of TMTCs and PMTs and further point to the importance of β-trefoil MIR domains for the glycosylation of S/T-rich multi-acceptor substrates.

Finally, the question remains whether our data could provide structural and functional explanations for mutations in the human POMT1/2 homologs that cause α-dystroglycanopathies (reviewed in *Endo, 2015*). Most of the missense mutations in POMT1/2 are found on the ER lumenal side, of which one third are within the MIR domains (*Figure 2*). When mapped onto the Pmt2-MIR structure, it is evident that most of them locate to the triangular cap region, although no obvious clustering can be observed (*Figure 8—figure supplement 2*). There are only a few mutations that directly affect site α (Q506 in Pmt2: $Q_{499}R$ mutation in POMT2 [*Østergaard et al., 2018*]) or site β (T419, V443 in Pmt2: $T_{414}M$, $V_{428}D$ in POMT1 [*Bello et al., 2012*; *Beltrán-Valero de Bernabé et al., 2002*]), while none occur in sites δ or δa. Interestingly, most of the point mutations concern non-solvent exposed polar residues involved in hydrogen-bonding networks, and thus their exchange might destabilize the MIR domains reflecting the effect of histidine mutations within Pmt2-MIR site α as measured here by nanoDSF. Since α-DG has a mucin-type *O*-glycosylation site in its central domain containing more than 40 S/T residues (*Gomez Toledo et al., 2012*), we deduce that in human POMT proteins, MIR domains use conserved mechanisms for *bona fide* substrate *O*-mannosylation, as described here for the Pmt1-Pmt2 complex. Sequence conservation suggests that those mechanisms are retained across PMT-MIR domains. In such a scenario, we speculate that Pmt1- and Pmt2-MIR domains interact in trans with the Pmt2- and Pmt1-TMDs and complement their respective active centers. Due to the asymmetry, both active centers are likely occupied sequentially rather than simultaneously by the S/T residue to be *O*-mannosylated within the acceptor polypeptide chain. In this manner, while the Pmt2-MIR domain holds the newly added mannose, mannosylation could proceed in the other active center at the level of the Pmt2-TMD and vice versa. Whether this mechanism holds also true for the Pmt4 homodimer remains open. Further studies are needed to derive mechanistic details e.g. by capturing PMT dimers in complex with bound donor and acceptor substrates or product peptides, or to challenge such a scenario and to finally gain insights into the mechanisms driving α-dystroglycanopathies.

# Materials and methods

**Key resources table**

| Reagent type (species) or resource | Designation | Source or reference | Identifiers | Additional information |
|---|---|---|---|---|
| Gene (*Saccharomyces cerevisiae*) | *PMT2* | *Saccharomyces* Genome Database (SGD) | YAL023C | |
| Gene (*S. cerevisiae*) | *PMT3* | SGD | YOR321W | |
| Gene (*S. cerevisiae*) | *PMT5* | SGD | YDL093W | |
| Gene (*S. cerevisiae*) | *HSP150* | SGD | YJL159W | |
| Gene (*S. cerevisiae*) | *SCW4* | SGD | YGR279C | |
| Strain, strain background (*S. cerevisiae*) | BY4741 | ***Brachmann et al., 1998*** | | MATa; his3-1; leu2-0; met15-0; ura3-0 |
| Strain, strain background (*S. cerevisiae*) | Y00385 (*pmt2Δ*) | EUROSCARF | | (BY4741) YAL023c::kanMX4 |
| Strain, strain background (*Escherichia coli*) | BL21(DE3) | Sigma-Aldrich | CMC0016 | Electro competent cells |
| Strain, strain background (*E. coli*) | SHuffle T7 Express | NEB | C3029JVIAL | Electro competent cells |
| Antibody | anti-HA (mouse, monoclonal) | Covance | #MMS-101R | WB (1:10000) |
| Recombinant DNA reagent | pAG1 (plasmid) | This paper | | Plasmid containing pYEP352 backbone and m*PMT2-HA* ($H_{362}A,H_{364}A$) |
| Recombinant DNA reagent | pAG2 (plasmid) | This paper | | Plasmid containing pYEP352 backbone and m*PMT2-HA* ($H_{428}A,H_{430}A$) |
| Recombinant DNA reagent | pAG4 (plasmid) | This paper | | Plasmid containing pYEP352 backbone and m*PMT2-HA* ($H_{362}A,H_{364}A,H_{428}A,H_{430}A$) |
| Recombinant DNA reagent | pAG9 (plasmid) | This paper | | Plasmid containing pYEP352 backbone and m*PMT2-HA* ($F_{516}A,K_{517}A$) |
| Recombinant DNA reagent | pYEP351a (plasmid) | Gift from V. Mrsa | | Plasmid containing pYEP351 backbone and $P_{GAL1}$-*SCW4*-HA |
| Sequence-based reagent | Forward primer used for construction of pAG1 and pAG4 | This paper | PCR primers | AGCTATACAAACTTATCCAGATGG |
| Sequence-based reagent | Reverse primer used for construction of pAG1 and pAG4 | This paper | PCR primers | GATGCCAATAGAGATCCTCCAAG |
| Sequence-based reagent | Forward primer used for construction of pAG2 and pAG4 | This paper | PCR primers | CGCTCCAGTTGCTGCACCAGTG |
| Sequence-based reagent | Reverse primer used for construction of pAG2 and pAG4 | This paper | PCR primers | GTAGCCAAGTTTCTGCCCGTGCTTTT |
| Sequence-based reagent | Forward primer used for construction of pAG9 | This paper | PCR primers | GCAAGGGACAAGAGGACCTGGTGG |
| Sequence-based reagent | Reverse primer used for construction of pAG9 | This paper | PCR primers | TGCTGGGTTTTTCATGCAGACAACCTC |
| Commercial assay or kit | Q5-site- directed Mutagenesis Kit | New England Biolabs (NEB) | E0554S | |

*Continued on next page*

*Continued*

| Reagent type (species) or resource | Designation | Source or reference | Identifiers | Additional information |
|---|---|---|---|---|
| Recombinant DNA reagent | pETHis (vector) | *Bogomolovas et al., 2009* | | |
| Sequence-based reagent | Forward primer used for construction of Pmt2-MIR | This paper | PCR primers | GCTTTCCATGGGCCCCC GTGACATTGCTCT |
| Sequence-based reagent | Reverse primer used for construction of Pmt2-MIR | This paper | PCR primers | GCTTTGGATCCTTATTCGGG TCTTGGTGGCAACCTT |
| Sequence-based reagent | Forward primer used for construction of Pmt3-MIR | This paper | PCR primers | GCTTTCCATGGGACCA CGTGATGTTGCTTTGG |
| Sequence-based reagent | Reverse primer used for construction of Pmt3-MIR | This paper | PCR primers | GCTTTGGATCCTTATTCT CCCTGTGGCAATCTTTCATTTTC |
| Sequence-based reagent | Forward primer used for construction of Pmt5-MIR | This paper | PCR primers | GCTTTCCATGGAGACTGT GGCAGAAGTTGCAG |
| Sequence-based reagent | Reverse primer used for construction of Pmt5-MIR | This paper | PCR primers | GCTTTCTCGAGTTACTCTGG ATTTGGCAAAGAAATTTCGTT |
| Commercial assay or kit | QuikChange | Agilent | | |
| Software | XDS | XDS package | | http://www.xds.mpimf-heidelberg.mpg.de |
| Software | AIMLESS | CCP4 suite | | http://www.mrc-lmb.cam.ac.uk/harry /pre/aimless.html |
| Software | COOT | Coot | | www.mrclmb.cam.ac.uk/personal/pemsley/coot/ |
| Software | PHENIX | Phenix suite | | https://www.phenix-online.org/ |
| Software | PyMol | Schroedinger | | http://www.pymol.org |
| Commercial assay or kit | RED-tris-NTA | NanoTemper | | http://www.nanotem-pertech.com |
| Software | Prometheus NT.48 | NanoTemper | PR.ThermContro | http://www.nanotempertech.com |
| Software | Monolith NT.115 | NanoTemper | MO.Affinity Analysis | http://www.nanotempertech.com |
| Software | Bruker Biospin | | NMR data acquisition and processing | https://www.bruker.com |
| Software | Sparky UCSF | | NMR spectra analysis | https://www.cgl.ucsf.edu/home/sparky/ |

## Cloning and expression of Pmt-MIR domains

The Pmt2-MIR domain (residues 338–539) was cloned using *Nco*I and *Bam*HI restriction sites, in frame with a cleavable (3C cleavage site) or non-cleavable N-terminal His6 tag in the pETHis vector (*Bogomolovas et al., 2009*). The cleavable construct was used to produce the protein for crystallization studies and the non-cleavable one to express and purifiy the protein for MST experiments. The Pmt2-MIR-$H_{362}A,H_{364}A$ mutant was generated using the QuikChange Lightning site-directed mutagenesis kit (Agilent Technologies). All constructs were expressed at 37°C in the *E. coli* T7 Shuffle strain and grown in standard TB medium supplemented with 0.4% (v/v) glycerol. At an $OD_{600}$ of 0.8, IPTG was added at a final concentration of 1 mM and cells were left to express over night at 22°C.

The Pmt3-MIR domain (residues 331–532) was cloned using *Nco*I and *Bam*HI restriction sites, in frame with an N-terminal DsbCin-His6-TEV sequence in the pETHis vector. Expression of the construct was done in *E. coli* Origami DE3 in auto-induction medium (*Studier, 2005*) for 20 hr at 24°C.

The Pmt5-MIR domain (residues 317–520) was cloned in frame with a non-cleavable N-terminal His6 tag, and the protein was expressed and purified as described for the Pmt2-MIR domain.

## Purification of Pmt-MIR domains

Cells were resuspended in a buffer containing 20 mM Tris-HCl (pH 7.0), 200 mM NaCl, 1 mM MgCl$_2$, and 20 mM imidazole supplemented with 1x Protease Inhibitor cocktail (Roche) and DNase I (Sigma-Aldrich). Cells were lysed through a microfluidizer, the lysate was cleared (48,000 x g, 4°C, 25 min) and the protein purified via HisTrap column (GE Healthcare). To release Pmt2-MIR, 3C protease was added to the protein bound to the column. For Pmt3-MIR, following imidazole elution TEV protease was added to remove the His-tag, while the His-tag was kept for the Pmt2-MIR domain, its mutant and Pmt5-MIR used for nanoDSF and MST analyses. TEV cleavage was performed over night at 4°C and the protease was removed by reloading on a HisTrap column. HisTrap eluates were concentrated and applied to Superdex 75 26/600 gel-filtration (GE Healthcare) pre-equilibrated in 20 mM Tris-HCl (pH 7.0) with either 150 mM NaCl for Pmt2-MIR, its mutant variant and Pmt5-MIR, or 200 mM NaCl for Pmt3-MIR. Fractions containing pure protein were pooled and concentrated to 13 mg/mL for Pmt2-MIR, 15 mg/mL for Pmt3-MIR, and 5 mg/mL for Pmt5-MIR for further studies.

## Crystallization and data collection

Crystals of Pmt2-MIR were grown in sitting drops (200 nL protein + 100 nL reservoir) equilibrated at 18°C against 100 μL reservoir containing 85 mM HEPES (pH 7.5), 1.7 M ammonium sulfate, 0.2 M magnesium acetate, 5.45 mM sodium acetate, 1.7% (v/v) polyethylene glycol 400% and 15% (v/v) glycerol. Crystals were directly harvested from the drop and plunged into liquid nitrogen. Diffraction data were recorded at 100 K at the European Synchrotron Radiation Facility (ESRF) on beamline ID30B. All datasets were integrated with XDS (*Kabsch, 2010*) and scaled with AIMLESS (*Evans and Murshudov, 2013*).

Crystals of Pmt3-MIR were grown accordingly over a reservoir containing 0.1 M MES (pH 6.0), 0.26 M calcium acetate and 15% (w/v) polyethylene glycol 8000. Crystals were cryo-protected by a quick soak in reservoir supplemented with 20% (v/v) glycerol immediately before snap-cryocooling by plunging into liquid nitrogen. Diffraction data were recorded at 100 K at the ESRF on beamline ID23-1 and data processed as for Pmt2-MIR.

## X-ray structure solution, refinement, and validation

For Pmt2-MIR, molecular replacement was performed with the PHENIX suite (*Adams et al., 2010*) using the *Arabidopsis thaliana* stromal cell-derived factor 2-like protein (SDF2, PDB-ID code 3MAL [*Schott et al., 2010*]) as a search model. Manual building was performed in COOT (*Emsley and Cowtan, 2004*) and refinement was performed with PHENIX. Validation was performed with MolProbity (*Chen et al., 2010*). For Pmt3-MIR, the Pmt2-MIR model was used as a search model and refinement and validation were done accordingly. Glycerol binding to site δ was identified in a dataset at 2.3 Å (Pmt2-MIR low, *Table 1*), that was not used for final refinement. Figures were prepared using PyMOL (DeLano Scientific).

## NMR measurements

NMR experiments were performed at a temperature of 298 K on Bruker spectrometers (600, 700, 800, 900 or 950 MHz) equipped with cryogenic triple-resonance probes. The Pmt2-MIR protein (0.1–0.4 mM) was expressed in *E. coli*, isotope labeled ($^{15}$N or $^{13}$C$^{15}$N) using M9 minimal media (supplemented with $^{15}$N-ammoniumchloride and $^{13}$C-D-glucose), purified (NiNTA affinity and gel-filtration on superdex 75) and measured in 20 mM HEPES buffer at pH 7.5 (containing 150 mM NaCl, 5 mM MgCl$_2$, and 10% D$_2$O) using 3 mm NMR tubes. The spectrometer was locked on D$_2$O and the NMR spectra were referenced using sodiumtrimethylsilylpropanesulfonate (DSS) as internal standard. For the backbone resonance assignment, a set of 3D triple-resonance experiments were collected including HNCO, HN(CA)CO, HNCA, HN(CO)CA, HNCACB, HN(CO)CACB and CC(CO)NH experiments on the $^{13}$C$^{15}$N labeled protein samples. Interactions were monitored by following the signals (to determine the amide chemical shift perturbations, CSPs) in a series of 1D $^1$H and 2D $^1$H$^{15}$N-HSQC experiments to an excess of the partner (glucose, glucose-1-phosphate, glucose-6-phosphate, methyl-α-D-glucoside, mannose, mannose-1-phosphate, mannose-6-phosphate, methyl-β-D-mannoside, YATAV, YAT(O-Man)AV, CYATAV, and CYAT(O-Man)AV). All spectra were processed using Topspin version 3.2 (Bruker Biospin) and analyzed using SPARKY version 3.114 (T. D. Goddard and D. G. Kneller, University of California, San Francisco).

## NanoDSF experiments to determine thermal stability

Pmt2-MIR, its mutant variant, and Pmt5-MIR were used at final concentrations of 25 µM and 50 µM in 150 mM NaCl and 20 mM Hepes (pH 7.5). The samples were loaded into UV capillaries and experiments performed with Prometheus NT.48 (NanoTemper Technologies). The temperature gradient was set to an increase of 1 ˚C/min with a range from 20˚C to 80˚C. The temperature-dependent change in tryptophan fluorescence at emission wavelength of 350 nm was monitored to measure protein unfolding. The melting temperature ($T_m$) was determined by detecting the maximum of the first derivative of the emission wavelength at 350 nm.

## MST measurements

Proteins were diluted to 400 nM in Phosphate Buffer Saline with 0.05% (v/v) Tween-20 and labeled with the His-tag label RED-tris-NTA (NanoTemper technologies) according to the manufacturer's protocol. For protein labeling equal volumes of 400 nM protein sample and 200 nM dye were mixed and the labeling reaction was let for 30 min at room temperature in the dark. Prior to usage, the protein was spun-down at 14,000 rpm for 10 min and kept in the dark on ice.

Meanwhile, sixteen 1:1 serial dilutions of the peptide were prepared in low binding profile tubes by adding an equal volume of the labeled protein. About 10 µL of the mixtures were transferred into glass capillaries, prior MST measurements on a Monolith NT.115 (NanoTemper technologies). Measures were carried out at 100% LED excitation power with 40 and 80% infrared-laser power (MST power). Experiments were done in triplicates. Averaged data points collected from the three independent measurements were fitted using a $K_D$-model as implemented in the MO Affinity analysis Software (NanoTemper Technologies).

## Yeast strains, culture conditions, and plasmids

The *S. cerevisiae* wild type BY4741 (MATa; his3-1; leu2-0; met15-0; ura3-0 [*Brachmann et al., 1998*]), isogenic mutant strains pmt2Δ (Y00385; YAL023c::kanMX4; EUROSCARF) and AGY15 (isogenic to pmt2Δ; with chromosomal HSP150 tagged by the 6xHA tag; see below), as well as wild type strain SEY6210 (MATα, his3-Δ200, leu2-3-112, lys2-801, trp1-Δ901, ura3-52, suc2-Δ9 [*Robinson et al., 1988*]) and the isogenic mutant *pmt2pmt4* (except pmt2::LEU2, pmt4::TRP1 [*Gentzsch and Tanner, 1996*]) were used. Yeast strains were grown in standard yeast extract-peptone-dextrose (YPD) or yeast nitrogen base selective medium (YNB) supplemented with the required amino acids. Yeast transformations were performed following the method of *Gietz et al., 1992*. Plasmids pVG80 (PMT2-HA [*Girrbach and Strahl, 2003*]), pJC16 (ER-GFP [*Castells-Ballester et al., 2019*]), pYM16 (*Janke et al., 2004*), YEp352 (*Hill et al., 1986*) and YEp351a (SCW4-HA; gift form V. Mrša) were used.

All pAG plasmids were generated using Q5 site-directed mutagenesis kit (New England Biolabs). For pAG1 ($H_{362}A$, $H_{364}A$), pAG2 ($H_{428}A$, $H_{430}A$), pAG3($C_{379}A$), pAG7 ($F_{516}A$), pAG8 ($K_{517}A$) and pAG9 ($F_{516}A$, $K_{517}A$) the plasmid pVG80 was used as a template, whereas for pAG4 ($H_{362}A$, $H_{364}A$, $H_{428}A$, $H_{430}A$) pAG1 was used. Plasmids pJS4 ($L_{361}A$), pJS5 ($L_{361}V$), pJS6 ($L_{427}A$), pAS151 ($Y_{380}A$), pAS152 ($D_{384}A$), pAS153 ($N_{386}A$) and pAS154 ($D_{384}A$, $N_{386}A$) were generated by site-directed mutagenesis using recombinant PCR as described previously (*Lommel et al., 2011*).

## Genomic tagging of HSP150

Hsp150 was genomically tagged at its C-terminal end with six copies of the HA epitope. Thereto, the 6x-HA-hphNT1 cassette was PCR amplified from plasmid pYM16 to create HSP150-6xHA. The resulting PCR products were transformed into *S. cerevisiae* strains BY4741, and pmt2Δ, and transformants selected on YPD plates containing hygromycin B (0.3 mg/mL, #10843555001, Roche).

## Spotting assay

Tenfold serial dilutions of mid-log phase *S. cerevisiae* cultures, starting from the concentration of $10^6$ cells/mL, were grown on appropriate selective YNB plates at indicated temperatures for up to 96 hr.

## Flow cytometry

*S. cerevisiae* cells expressing ER-GFP from plasmid pJC16 were grown to the mid-log phase, harvested and resuspended in PBS buffer to a concentration of $10^6$ cells/mL. Cells were gently

sonicated, and the GFP fluorescence intensity of 20,000 events was quantified by flow cytometry using BD FACSCanto II (BD Biosciences).

## Isolation of cell extracts, total membranes, and PMT substrate proteins

Isolation of cell extracts and total membranes from baker´s yeast was performed as described previously (*Castells-Ballester et al., 2019*). Scw4-HA was isolated following the protocol described by *Grbavac et al., 2017*. For the isolation of Hsp150-HA mid-log phase yeast cells were harvested (10 $OD_{600}$) and resuspended in 50 µL of appropriate growth medium. After 2 hr of incubation at 37°C, 40 µL of supernatant was collected and subjected to further analysis.

## SDS-PAGE and immunoblot

Protein samples were denatured in 1x SDS-sample buffer for 10 min at 70°C (for detection of Pmt2 and ER-GFP) or 3 min at 95°C (for detection of Hsp150), resolved on glycine polyacrylamide gels, and transferred to nitrocellulose membrane. Blots were incubated with primary antibodies anti-HA (from mouse; monoclonal; 1:10000; #MMS-101R; Covance), anti-Sec61 (from rabbit; polyclonal; 1:2500; gift from Karin Römisch), anti-GFP (from rabbit; polyclonal; 1:2500 #A6455; Thermo Fisher Scientific) or anti-G6PDH (from rabbit; polyclonal; 1:2500; #A9521; Sigma-Aldrich). Secondary antibodies horseradish peroxidase-conjugated anti-mouse (from rabbit; polyclonal; 1:5000; #A9044; Sigma-Aldrich) or anti-rabbit (from goat; polyclonal; 1:5000; #A6154; Sigma-Aldrich) were used. Protein-antibody complexes were visualized by enhanced chemiluminescence (Amersham ECL Detection Reagents; GE Healthcare).

## Acknowledgements

We thank Vladimir Mrša for the gift of plasmid YEp351a, and Karin Römisch for the gift of Sec61 directed antibodies. We are very grateful to Karine Lapouge for excellent scientific support, Anke Metschies and Elke Herwig for excellent technical support, Jessika Sonnabend for initial cloning and characterization of Pmt2 leucine mutants, Zuzanna Koskova for helping in MST measurements, Daniela Bausewein for many helpful discussions throughout this project, and Melanie McDowell for careful proofreading of the manuscript. We thank the Flow Cytometry and FACS Core Facility (ZMBH, Heidelberg University; Germany) for excellent technical support. We thank Jürgen Kopp and Claudia Siegmann from the BZH/Cluster of Excellence: CellNetworks crystallization platform and acknowledge access to the beamlines at the European Synchrotron Radiation Facility (ESRF) in Grenoble and the support of the beamline scientists. We acknowledge the data storage service SDS@hd supported by the Ministry of Science, Research and the Arts Baden-Württemberg (MWK) and the German Research Foundation (DFG) through grant INST 35/1314–1 and INST 35/1503–1 FUGG. IS is an investigator of the Cluster of Excellence: CellNetworks. This work was funded by the German Research Foundation (DFG) through Project-ID 201348542-SFB1036 (TP11 to SS and TP22 to IS), Forschungsgruppe FOR2509 (SCHW701/20-1 to HS; SI586/8-1 to IS; STR 443/5–1 to SS), and the Leibniz program (SI 586/6–1) to I.S. Work at BMRZ is supported by the state of Hessen.

## Additional information

### Funding

| Funder | Grant reference number | Author |
| --- | --- | --- |
| German Research Foundation | FOR2509 SI586/8-1 | Irmgard Sinning |
| German Research Foundation | 201348542-SFB1036 TP11 | Sabine Strahl |
| German Research Foundation | FOR2509 STR443/5-1 | Sabine Strahl |
| German Research Foundation | FOR2509 SCHW701/20-1 | Harald Schwalbe |
| German Research Foundation | Leibniz Program SI586/6-1 | Irmgard Sinning |
| German Research Foundation | 201348542-SFB1036 TP22 | Irmgard Sinning |

The funders had no role in study design, data collection and interpretation, or the decision to submit the work for publication.

## Author contributions

Antonella Chiapparino, Data curation, Investigation, Visualization, Methodology, Writing - original draft, Writing - review and editing; Antonija Grbavac, Data curation, Formal analysis, Validation, Investigation, Visualization, Methodology, Writing - original draft; Hendrik RA Jonker, Krishna Saxena, Data curation, Validation, Investigation, Methodology, Writing - original draft; Yvonne Hackmann, Formal analysis, Investigation, Writing - original draft; Sofia Mortensen, Ewa Zatorska, Data curation, Investigation; Andrea Schott, Data curation, Validation, Investigation; Gunter Stier, Data curation, Investigation, Methodology; Klemens Wild, Supervision, Validation, Investigation, Visualization, Methodology, Writing - original draft, Writing - review and editing; Harald Schwalbe, Conceptualization, Supervision, Funding acquisition, Validation, Methodology, Writing - original draft, Project administration; Sabine Strahl, Conceptualization, Supervision, Funding acquisition, Validation, Methodology, Writing - original draft, Project administration, Writing - review and editing; Irmgard Sinning, Conceptualization, Resources, Supervision, Funding acquisition, Validation, Methodology, Writing - original draft, Project administration, Writing - review and editing

## Author ORCIDs

Hendrik RA Jonker (iD) https://orcid.org/0000-0002-5582-3931
Sofia Mortensen (iD) https://orcid.org/0000-0003-1631-6651
Klemens Wild (iD) https://orcid.org/0000-0001-9733-8187
Harald Schwalbe (iD) http://orcid.org/0000-0001-5693-7909
Sabine Strahl (iD) https://orcid.org/0000-0002-4024-0741
Irmgard Sinning (iD) https://orcid.org/0000-0001-9127-4477

## Decision letter and Author response

Decision letter https://doi.org/10.7554/eLife.61189.sa1
Author response https://doi.org/10.7554/eLife.61189.sa2

# Additional files

## Supplementary files

- Transparent reporting form

## Data availability

The atomic coordinates and structure factors have been deposited in the PDB with accession ID 6ZQP (Pmt2-MIR) and ID 6ZQQ (Pmt3-MIR).

The following datasets were generated:

| Author(s) | Year | Dataset title | Dataset URL | Database and Identifier |
|---|---|---|---|---|
| Wild K, Chiapparino A, Hackmann Y, Mortensen S, Sinning I | 2020 | Structure of the Pmt2-MIR domain with bound ligands | https://www.rcsb.org/structure/6ZQP | RCSB Protein Data Bank, 6ZQP |
| Wild K, Chiapparino A, Hackmann Y, Mortensen S, Sinning I | 2020 | Structure of the Pmt2-MIR domain with bound ligands | https://www.rcsb.org/structure/6ZQQ | RCSB Protein Data Bank, 6ZQQ |

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
