## [Decision Letter]

**Acceptance summary:**

The manuscript presents more clearly the major structural findings, the mechanistic implications (to the extent that they are understood at this point), and relationships to other glycosyltransferase families.

**Decision letter after peer review:**

Thank you for submitting your article "Functional implications of MIR domains in protein O-mannosylation" for consideration by *eLife*. Your article has been reviewed by three peer reviewers, one of whom is a Guest Reviewing Editors, and the evaluation has been overseen by David Ron as the Senior Editor. The reviewers have opted to remain anonymous.

The reviewers have discussed the reviews with one another and the Reviewing Editor has drafted this decision to help you prepare a revised submission.

As the editors have judged that your manuscript is of interest, but as described below, substantial changes are required before it is published, we would like to draw your attention to changes in our revision policy that we have made in response to COVID-19 (https://elifesciences.org/articles/57162). First, because many researchers have temporarily lost access to the labs, we will give authors as much time as they need to submit revised manuscripts. We are also offering, if you choose, to post the manuscript to bioRxiv (if it is not already there) along with this decision letter and a formal designation that the manuscript is "in revision at *eLife*". Please let us know if you would like to pursue this option. (If your work is more suitable for medRxiv, you will need to post the preprint yourself, as the mechanisms for us to do so are still in development.)

Summary:

The manuscript describes the structures and sugar-binding properties of protein O-mannosyltransferase (Pmt) MIR domains. At least three valuable and justified insights are presented. First, mannose-modified peptides are bound by the MIR domain, and the binding site is identified. Second, requirements for mannosylation of Ser/Thr-rich substrates can be distinguished by mutagenesis from requirements for activity on mis-folded substrates lacking Ser/Thr-rich regions. Third, a comparison of Pmt enzymes with GalNAc transferases suggests mechanistic commonalities. A structural comparison with homologous MIR domains from inositol triphosphate receptors and ryanodine receptors is also described. These studies complement the recent report of the cryo-EM structure of the yeast Pmt1/Pmt2 complex, which also presented an MIR domain crystal structure and noted the similarity to sugar-binding proteins but did not further explore the implications of these observations.

The reviewers had mixed opinions on whether the novelty of the work or the mechanistic advances in light of the Pmt1/Pmt2 cryo-EM structure are sufficient to warrant publication in a general biology journal. Nevertheless, the efforts of the authors to experimentally validate their claims and also to place them in a broader context within the field of glycosyltransferases and β-trefoil containing proteins was appreciated. Consequently, it was decided to invite the authors to present a more concise and substantially revised version of the manuscript for re-evaluation, taking into consideration the reviewers' general and specific comments.

Essential revisions:

Attention should be paid to the suggestion for shortening and reorganization offered by reviewer #1.

Reviewer #1:

This manuscript picks up very appropriately where Bai et al., 2019 left off with their cryo-EM structure of the yeast Pmt1-Pmt2 mannosyltransferase complex. Bai et al. wrote at the end of their paper, "Based on this structural homology and the fact that the MIR domain is located just above the Dol-P-Man entry or product-release path, we suggest that it may play a role in recognition of the mannose moiety of the donor substrate. More studies are needed, however, to determine the function of this domain." Chiapparino and colleagues now make a significant advance in such studies.

The results in Chiapparino et al. include the following: crystal structures of MIR domains, suggestion of ligand-binding sites, comparison with MIR domains in other ER-localized proteins, comparison to other carbohydrate-binding trefoil fold structures, experimental analysis of carbohydrate and peptide binding, and a mutational analysis of MIR domain function in Pmt1-Pmt2 in vivo.

While overall the findings are valuable to report, the manuscript should be drastically reorganized and the presentation should be more focused. The authors should aim to shorten it to about 2/3 the current length. Rather than emphasize the "integrated structural biology" approach, the authors should focus on what experiments and analyses are truly valuable, and devote time to presenting them in as effective (and concise) a manner as possible. This reviewer felt that the story only really got going on in the subsection “Analysis of the Pmt2-MIR domain as a CBM” and wondered why so much text (subsections “PMT-MIR domain structures”, “The Pmt2-MIR domain contains three putative ligand-binding sites” and “The Pmt2-MIR domain has unique features within the MIR family”) was wasted to get to something that was essentially already known – that β-trefoil folds are recognized carbohydrate binding proteins and have recognized binding sites. The authors should summarize their structural studies as briefly as possible in the context of what is known already about carbohydrate binding in other trefoil CBMs and then focus on their own binding and functional experiments. To clarify, the order should be:

1) MIR domains are β-trefoils, a subset of CBMs are β-trefoils, and the structures of Pmt-MIR domains are consistent with their being CBMs;

2) NMR and MST measurements;

3) Comparison with GALNTs and analysis of the findings in the context of the structure and mechanism of the full-length Pmt1-Pmt2 complex.

On a side note, the discovery of domains that have evolved to provide related functions in different contexts brings unity and order to our understanding of nature. This process should be accompanied by unification of nomenclature. This reviewer was surprised and disappointed after reading about sites I, II, and III to find out that carbohydrate binding sites were already called α, β, and γ in other β-trefoils. (Actually, in Boraston et al., α, β, and γ refer to the subdomains and not the binding sites: "The three β-trefoil subdomains of SlCBM13 are labelled as α, β and γ…") If the authors were to discuss their structural work in terms of what is already known about β-trefoil CBMs as suggested above, they could use the conventional nomenclature (whatever it actually is) from the outset. For example, they could use the α, β, and γ designations and then introduce a putative δ site (their site III).

Reviewer #2:

A common post-translational modification in the endoplasmic reticulum is O-mannosylation or the transfer of a mannose carbohydrate to Ser/Thr residues of secretory pathway client proteins. In yeast there are seven protein O-mannosyltransferases (PMT1-7) while in mammals where they are termed POMT, there are only two (POMT1 and 2). This modification has functional implications especially for proteins in the cell wall (or the plasma membrane or extracellular matrix of mammalian cells). In yeast, it has also been shown to be used to signal proteins for degradation (UPOM; unfolded protein O-mannosylation). The Li lab recently solved the cryo-EM structure for the PMT1/PMT2 heterodimer. This interesting structure mainly focusses on the transmembrane domains and how they assemble and orientate the active site that binds the dolichol-monophosphate activated mannose substrate. In this important study, they also solved the structure of the luminal soluble MIR domain found between transmembrane domains 7 and 8. As this domain is the most significant luminal exposed part of the protein, it is expected to be involved in substrate recognition (especially for soluble substrates) though the Bai et al., study does not focus on this aspect. The role of the MIR domain in the PMTs is the focus on the current manuscript. Here, Chaipparino et al., have solved the X-ray crystal structure of the MIR domains from PMT2 and PMT3. Analysis of these β-trefoil structures and their comparison to similar known structures suggest that these MIR domains might be involved in carbohydrate binding (mannosylated peptide or dolichol-P-mannose). NMR and microscale thermophoresis analysis are supportive for a role in binding mannose modified peptides (product binding) but not the dolichol-P-mannose substrate, unmodified acceptor peptides or unfolded substrates (substrate recognition for UPOM).

Overall while the approach is suitable and experimental results are of high quality (MIR structure in 1.6 Å resolution). In short, I am not convinced that the insight uncovered significantly advances the mechanism of PMT O-mannosylation sufficiently to interest the broad readership of *eLife*.

If the MIR domain is not involved in initial substrate recognition/orientation, how are soluble substrates to be mannosylated recognized?

Compare the MIR domains of the UPOM PMTs (PMT_1/2_) with that of the MIR domains that modify properly folded Ser/Thr rich proteins (PMT4/4).

Figure 4A models the MIR domain on to the Bai et al., transmembrane regions of PMT_1/2_ and with the IP3R and RyR MIR domains. This should be broken up to first show the MIR domain solely on the PMT transmembrane region, followed by comparison of MIR domains from PMT, IP3R and RyR.

Reviewer #3:

The paper describes structural studies of two MIR domains involved

in mannosylation of proteins in the ER, and functional studies of mutants. Although outside my field I found the paper interesting, thorough and worthy of publication.

---

## [Author Response]

Essential revisions:Attention should be paid to the suggestion for shortening and reorganization offered by reviewer #1.Reviewer #1:[…] While overall the findings are valuable to report, the manuscript should be drastically reorganized and the presentation should be more focused. The authors should aim to shorten it to about 2/3 the current length. Rather than emphasize the "integrated structural biology" approach, the authors should focus on what experiments and analyses are truly valuable, and devote time to presenting them in as effective (and concise) a manner as possible. This reviewer felt that the story only really got going on in the subsection “Analysis of the Pmt2-MIR domain as a CBM” and wondered why so much text (subsections “PMT-MIR domain structures”, “The Pmt2-MIR domain contains three putative ligand-binding sites” and “The Pmt2-MIR domain has unique features within the MIR family”) was wasted to get to something that was essentially already known – that β-trefoil folds are recognized carbohydrate binding proteins and have recognized binding sites. The authors should summarize their structural studies as briefly as possible in the context of what is known already about carbohydrate binding in other trefoil CBMs and then focus on their own binding and functional experiments. To clarify, the order should be:1) MIR domains are β-trefoils, a subset of CBMs are β-trefoils, and the structures of Pmt-MIR domains are consistent with their being CBMs;2) NMR and MST measurements;3) Comparison with GALNTs and analysis of the findings in the context of the structure and mechanism of the full-length Pmt1-Pmt2 complex.

We thank the reviewer for the positive evaluation and all the valuable suggestions to improve our manuscript. Following this advice, we have made an effort to focus the manuscript as well as to shorten it considerably. It now strictly follows the order suggested by the reviewer.

In the previous version, we provided a detailed description of the β-trefoil fold as we found quite some confusion in how definitions are used in the literature. We wanted to get this sorted. In the revised version, we shortened this part considerably (by about a third). Further, we cut the MIR domain comparison significantly (no separate chapter anymore) and took the respective Figure 4 out of the main text. Finally, the mutational data concerning the protein core of Pmt1 has been taken out (including Supplementary Figure 1).

On a side note, the discovery of domains that have evolved to provide related functions in different contexts brings unity and order to our understanding of nature. This process should be accompanied by unification of nomenclature. This reviewer was surprised and disappointed after reading about sites I, II, and III to find out that carbohydrate binding sites were already called α, β, and γ in other β-trefoils. (Actually, in Boraston et al., α, β, and γ refer to the subdomains and not the binding sites: "The three β-trefoil subdomains of SlCBM13 are labelled as α, β and γ…") If the authors were to discuss their structural work in terms of what is already known about β-trefoil CBMs as suggested above, they could use the conventional nomenclature (whatever it actually is) from the outset. For example, they could use the α, β, and γ designations and then introduce a putative δ site (their site III).

We apologize for the confusion. Our intention was not to introduce new nomenclature, but our nomenclature was introduced in an unbiased, structure-based approach and was later on found to (largely) coincide with the CBM definitions (which however are not consistent in the literature). To resolve this issue – we now comply with the CBM nomenclature and refer to sites α to δ as suggested.

Reviewer #2:[…] Overall while the approach is suitable and experimental results are of high quality (MIR structure in 1.6 Å resolution). In short, I am not convinced that the insight uncovered significantly advances the mechanism of PMT O-mannosylation sufficiently to interest the broad readership of eLife.

We thank the reviewer for summarizing our main findings. We however respectfully disagree with the reviewer as we are convinced that our study does significantly contribute to our understanding of the role of MIR domains in protein-O-mannosylation.

If the MIR domain is not involved in initial substrate recognition/orientation, how are soluble substrates to be mannosylated recognized?

This is a very important point – but unfortunately not clear yet. As we deduce from the recent Pmt1-Pmt2 cryo-EM structure (Bai et al., 2019), although it was not described there, recognition is likely to occur in the interface between the TMDs and MIR domains. From the MIR side, it seems that the conserved site δa could be involved in substrate recognition, as stated in the Discussion:

“Finally, the high conservation of site δa (next to site δ) within all PMT-MIR domains still remains enigmatic, but due to its position in close proximity to the active site it might be involved in substrate-peptide recognition.”

Compare the MIR domains of the UPOM PMTs (PMT_1/2_) with that of the MIR domains that modify properly folded Ser/Thr rich proteins (PMT4/4).

This is not possible yet as a structure of the Pmt4-MIR domain is not available. Pmt4-MIR (as POMT1) contains e.g. another 7 residue insertion of unknown function (see Figure 2).

Figure 4A models the MIR domain on to the Bai et al., transmembrane regions of PMT_1/2_ and with the IP3R and RyR MIR domains. This should be broken up to first show the MIR domain solely on the PMT transmembrane region, followed by comparison of MIR domains from PMT, IP3R and RyR.

The MIR domain comparison has been significantly reduced now (as requested also by reviewer #1) and only the panels of MIR domain superposition have been kept (now in Figure 1—figure supplement 2).